# Discrete hippocampal projections are differentially regulated by parvalbumin and somatostatin interneurons

Daniel J. Lodge [1,2], Hannah B. Reiley [1,2], Angela M. Boley [1,2] & Jennifer J. Donegan [1,3] ✉

People with schizophrenia show hyperactivity in the ventral hippocampus (vHipp) and we have previously demonstrated distinct behavioral roles for vHipp cell populations. Here, we test the hypothesis that parvalbumin (PV) and somatostatin (SST) interneurons differentially innervate and regulate hippocampal pyramidal neurons based on their projection target. First, we use eGRASP to show that PV-positive interneurons form a similar number of synaptic connections with pyramidal cells regardless of their projection target while SST-positive interneurons preferentially target nucleus accumbens (NAc) projections. To determine if these anatomical differences result in functional changes, we used in vivo opto-electrophysiology to show that SST cells also preferentially regulate the activity of NAc-projecting cells. These results suggest vHipp interneurons differentially regulate that vHipp neurons that project to the medial prefrontal cortex (mPFC) and NAc. Characterization of these cell populations may provide potential molecular targets for the treatment schizophrenia and other psychiatric disorders associated with vHipp dysfunction.

The hippocampus has long been recognized for its role in memory and spatial navigation[1–4]. However, over the past few decades it has become apparent that the hippocampus also mediates reward-seeking[5], anxiety-related behaviors[6], and the neuroendocrine response to stress[6,7], leading many to explore the role of the hippocampus in psychiatric disorders. In patients with schizophrenia, volume loss is consistently observed in the hippocampus[8,9]. The decrease in volume is accompanied by an increase in hippocampal activity, which is correlated with the severity of positive symptoms of schizophrenia[10–13]. In rodents, hippocampal hyperactivity has been observed across multiple models used to study psychosis[14–18] and has been shown to underlie schizophrenia-related deficits, including dopamine cell hyperactivity, impaired sensory motor gating, decreased social interaction, and cognitive inflexibility[19–21].

The hippocampus is a highly organized structure, which has been divided into multiple subregions including the dentate gyrus (DG), Cornu Ammonis (CA)1, CA2, CA3, and subiculum. More recent work has also demonstrated variations along the dorsal-ventral axis (posterior-anterior in humans)[1,22,23] with differences in connectivity[24,25], electrophysiological properties[26–28], gene expression[29–32], and behavioral regulation[2,7] observed between the dorsal and ventral hippocampus. For example, more ventral (anterior) regions of the hippocampus are highly connected to limbic circuitry[24,25] and have been shown to regulate motivated and emotional behaviors[6,33,34]. However, even within the ventral hippocampus (vHipp), further subdivisions can be made[29,31,32,35,36]. We have shown previously that unique projections from the vHipp differentially regulate schizophrenia-like behaviors in a rodent model. Specifically, vHipp →nucleus accumbens

[1]Department of Pharmacology and Center for Biomedical Neuroscience, University of Texas Health Science Center, San Antonio, TX 78229, USA. [2]South Texas Veterans Health Care System, Audie L. Murphy Division, San Antonio, TX, USA. [3]Department of Psychiatry and Behavioral Sciences and Center for Early Life Adversity, Department of Neuroscience, Dell Medical School at the University of Texas at Austin, Austin, TX 78712, USA. ✉e-mail: jennifer.donegan@austin.utexas.edu

(NAc) projections regulate the activity of dopamine cells in the ventral tegmental area and sensory motor gating while vHipp→medial prefrontal cortex (mPFC) projections regulate social interaction, cognitive function[37], and the antidepressant response to ketamine[38].

Similar functional distinctions between unique cell types have been demonstrated when using a cell transplantation approach to examine the role of discrete interneuron subtypes in symptoms associated with schizophrenia. Specifically, we demonstrated previously that vHipp transplantation of somatostatin interneurons (SST) into a rodent model, rescued deficits associated with negative and cognitive symptoms while vHipp transplantation of parvalbumin (PV) interneurons reversed deficits across all symptom domains[19]. This led to an intriguing hypothesis that PV and SST interneurons may differentially innervate hippocampal pyramidal neurons based on their projection target.

Therefore, in the current experiments, we aimed to characterize microcircuits within the vHipp based on their projection target (either NAc or mPFC). We first used retrograde viral tracing to demonstrate that NAc- and mPFC-projecting pyramidal cells represent two unique and anatomically segregated cell populations. Interestingly, these two cell populations also display unique transcriptional profiles, determined by flow-cytometry and RNASeq. To examine discrete microcircuits regulating NAc- or mPFC- projecting neurons in the vHipp, enhanced GFP recombination across synaptic partners (eGRASP), immuno-electron microscopy, and mono-synaptic rabies virus tracing were used. Finally, opto-electrophysiology, combined with optogenetics, was used to confirm that the activity of NAc- and mPFC-projecting pyramidal cells are differentially regulated by unique interneuron subtypes. Together, these experiments confirm that unique microcircuits within the vHipp differentially regulate pyramidal cell populations that are relevant to discrete schizophrenia-like behaviors.

## Results

### vHipp projections to the mPFC and NAc are anatomically segregated

To determine the anatomical location of vHipp projection neurons, retrograde viruses expressing either myrGFP or myrScarlet were injected into either the mPFC or NAc (Fig. 1A) and cell bodies were identified in the vHipp. Approximately 43% of the labeled cells project to the NAc, while 56% project to the mPFC. Only 1% of cells projected to both regions (Fig. 1B; NAc = $5.39 \pm 1.06$ cells per slice, mPFC = $6.69 \pm 1.09$ cells per slice, Both = $0.02 \pm 0.12$ cells per slice). The anatomical location of the mPFC- or NAc-projecting cells was mapped onto a schematic of the vHipp (Fig. 1C), demonstrating that the majority of mPFC-projecting cell bodies are localized to the ventral CA1, while the majority of NAc-projecting cell bodies are found in the ventral subiculum. A representative image of the vHipp projections to the mPFC and NAc is shown in Fig. 1D. These results demonstrate that NAc- and mPFC-projecting neurons are made up of distinct and anatomically segregated populations of cells.

### vHipp projections to the mPFC and NAc have unique transcriptional signatures

To determine the transcriptional profile of mPFC- and NAc-projecting neurons, RNA Sequencing was performed on retrogradely labeled neurons separated by flow cytometry (Fig. 2A). NAc-projecting neurons made up $3.22 \pm 1.036\%$ of cells sorted while mPFC-projecting neurons made up $3.610 \pm 1.84\%$ of cells sorted. RNASeq identified 99 genes that were differentially expressed between NAc- and

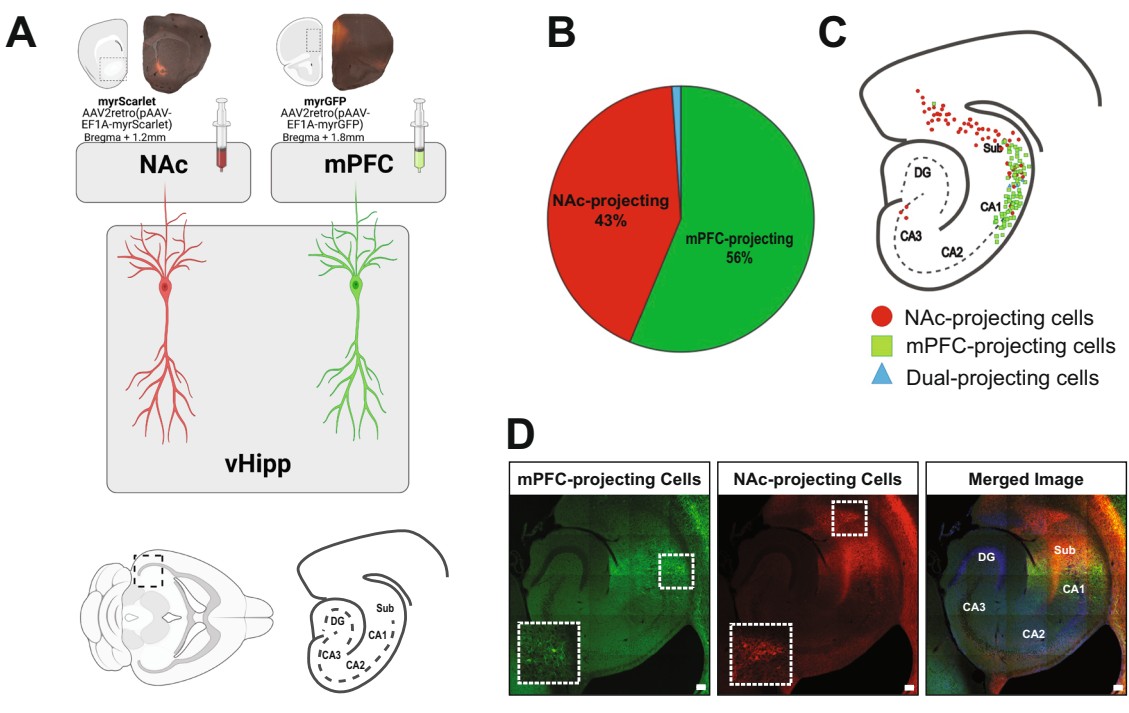

**Fig. 1 | Ventral hippocampal (vHipp) projections to the medial prefrontal cortex (mPFC) and nucleus accumbens (NAc) are anatomically segregated.**
**A** To identify vHipp projection neurons, retrograde viruses expressing either myristoylated Scarlet (myrScarlet) or myristoylated Green Fluorescent Protein (myrGFP) were injected into the NAc or mPFC. **B** The total number of fluorescently labeled cells in the vHipp were counted and the percentage projecting to each region was determined. **C** The anatomical location of each cell was mapped onto a cartoon of the vHipp. A representative image of the ventral hippocampus is depicted in (**D**). mPFC-projecting cells are depicted in green, NAc-projecting cells are shown in red. Scale bar is 100 microns. $n = 2$–4 sections each from 4 mice per group. Source data are provided as a Source Data file.

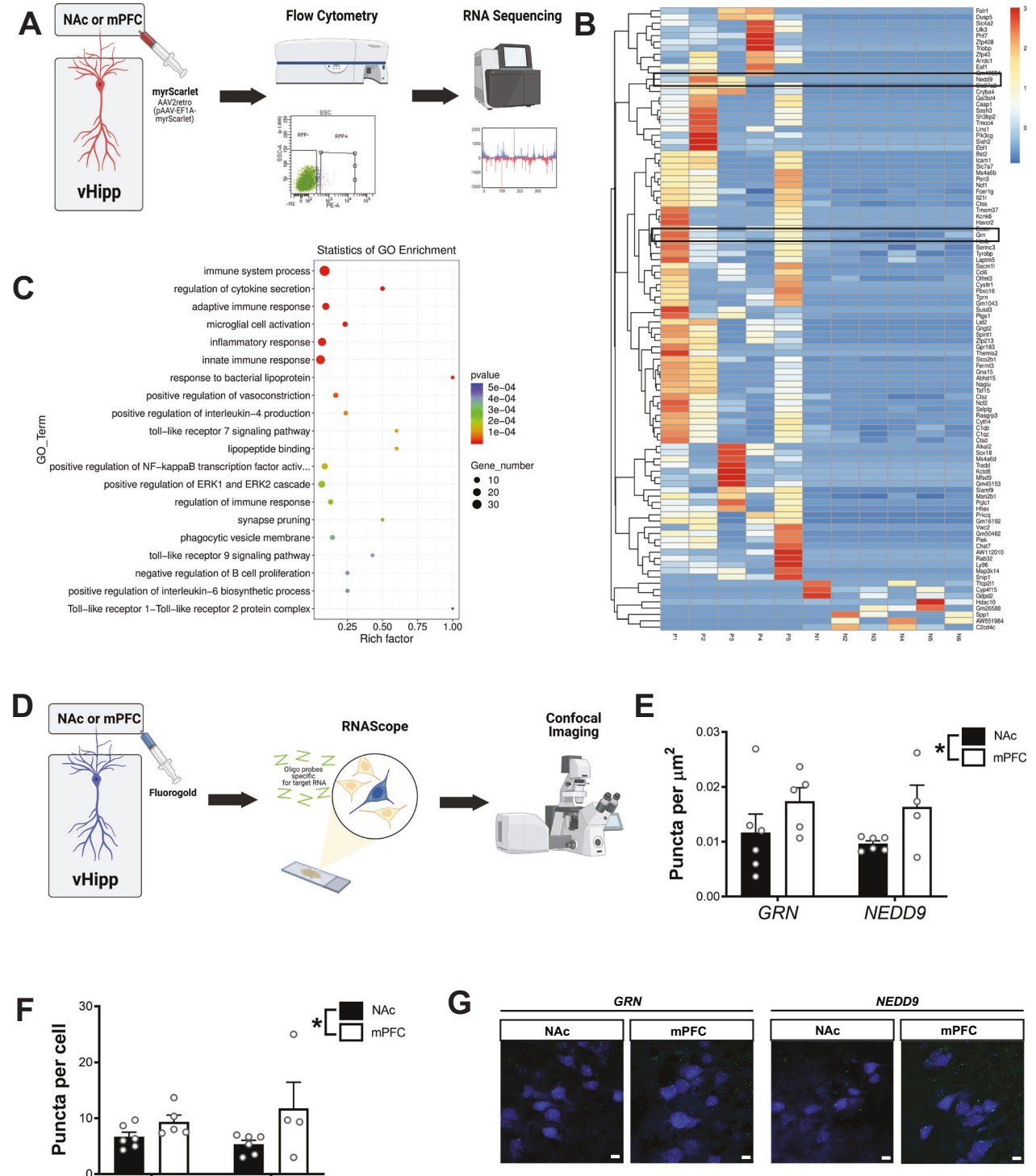

**Fig. 2 | Ventral hippocampal (vHipp) projections to the medial prefrontal cortex (mPFC) and nucleus accumbens (NAc) have unique transcriptional signatures. A** To determine gene expression in vHipp projection neurons, a retrograde virus expressing myrScarlet was injected into the mPFC or NAc. Scarlet-positive cells in the vHipp were isolated by flow cytometry and RNASequencing (RNASeq) was used to measure gene expression. **B** RNASeq identified 92 genes that were differentially expressed between mPFC- and NAc-projecting cells. *GRN* and *NEDD9* are outlined in black. **C** Gene ontology was used to determine enriched pathways. *n* = 5 (mPFC-projecting cells) or 6 (NAc-projecting cells) per group. Scale bar is 50 μm. **D** To confirm RNASeq analysis, fluorogold was injected into the mPFC or NAc and RNAScope was used to measure expression of *GRN* and *NEDD9* using

RNAScope. **E** In line with the RNASeq results, *GRN* and *NEDD9* were increased in mPFC-projecting cells compared to those that project to the NAc. Data were analyzed by Two-way ANOVA, *p* = 0.0369. **F** When total number of puncta per cell were analyzed, *GRN* and *NEDD9* were increased in mPFC-projecting cells compared to those that project to the NAc. Data were analyzed by Two-way ANOVA, *p* = 0.0319. Representative images are shown in (**G**). * signifies a main effect of projection target. Scale bar is 10 microns. *n* = 3 cells each from 4 (mPFC-projecting cells, *NEDD9*), 5 (mPFC-projecting cells, *GRN*), or 6 mice (NAc-projecting cells, *GRN* or *NEDD9*) per group. All data are presented as mean values +/- SEM. Source data are provided as a Source Data file.

mPFC- projecting neurons (Fig. 2B). Of these genes, 92 showed significantly greater expression in mPFC- compared to NAc-projecting neurons. Gene Ontology (GO) was used to identify the biological processes associated with these differentially expressed genes (Fig. 2C). The majority of differentially expressed genes were involved in immune signaling, including inflammatory response, innate immune response, and immune system processes. However, GO enrichment also identified synapse pruning and multiple genes associated with synaptic plasticity, including the complement pathway proteins C1qa, C1qb, and C1qc, were differentially expressed between the two neuron populations. Importantly, neither RNA Integrity Number (RIN) values nor mapped read counts differed between the two projection pathways (NAc-projecting cells RIN = $5.067 \pm 0.619$, mPFC-projecting cells RIN = $5.160 \pm 1.204$; NAc-projecting cells mapped reads = 21816520 $(71.672\%) \pm 1297102$, mPFC-projecting cells mapped reads = 23663120 $(75.514\%) \pm 690364$). A list of differentially expressed genes is included in Supplementary Data 1.

To confirm that the gene expression differences could not be explained by differences between anatomical subregions, we used the Cytosplore Viewer tool (https://viewer.cytosplore.org)[39–41] to compare expression of the identified genes between CA1 and subiculum. Of the 99 genes that were differentially expressed between mPFC- and NAc-projecting neurons, only 12 also showed differential expression between hippocampal CA1 and subiculum (Supplementary Data 2). This suggests that differential gene expression observed in mPFC- and NAc- projecting neurons is not solely a result of their anatomical location.

To confirm the RNASeq results, RNAScope was used to measure the expression of two genes implicated in synaptic plasticity, *NEDD9* and *GRN*, in mPFC- or NAc-projecting neurons (Fig. 2D). In line with the RNA Sequencing results, we found a main effect of projection target (Fig. 2E; Two-way ANOVA $F_{(1,17)} = 5.13$, $p < 0.05$). Specifically, mPFC-projecting neurons had greater *GRN* and *NEDD9* expression than vHipp pyramidal cells that project to the NAc (*GRN*: NAc = $0.012 \pm 0.003$ puncta per μm, mPFC = $0.017 \pm 0.002$ puncta per μm; *NEDD9*: NAc = $0.010 \pm 0.001$ puncta per μm, mPFC = $0.016 \pm 0.004$ puncta per μm). The same effect was observed when the total number of puncta per cell was analyzed (Fig. 2F; *GRN*: NAc = $6.67 \pm 0.85$ puncta per cell, mPFC = $9.37 \pm 1.20$ puncta per cell; *NEDD9*: NAc = $5.36 \pm 0.71$ puncta per cell, mPFC = $11.75 \pm 4.68$ puncta per cell; Two-way ANOVA F $_{(1,17)} = 5.46$, $p < 0.05$). Representative images are shown in Fig. 2G. These results suggest NAc- and mPFC-projecting neurons have distinct transcriptional signatures.

## vHipp projections to the mPFC and NAc are differentially innervated by local interneurons

Next, we aimed to understand the microcircuits associated with discrete vHipp projection neurons. Therefore, we used eGRASP to label synaptic connections between inhibitory interneuron subtypes (parvalbumin (PV)- or somatostatin (SST)-positive interneurons) and pyramidal cells that project to the NAc or mPFC (Fig. 3A, B). We found an interaction between interneuron subtype and projection target (Fig. 3C; Shapiro–Wilk Normality Test P = 0.27; Two-way ANOVA $F_{(1,15)} = 8.034$, $p < 0.05$; $n = 3$ cells each from 3–7 mice). Specifically, PV-positive interneurons form a similar number of synapses on vHipp pyramidal cells that project to the NAc and mPFC (PV:NAc vs PV:mPFC: Holm-Sidak $t = 0.98$, $p > 0.05$; NAc = $2.89 \pm 1.26$ synapses per μm, mPFC = $4.2 \pm 1.44$ synapses per μm). SST-positive interneurons, however, preferentially synapse on vHipp neurons projecting to the NAc (SST:NAc vs SST:mPFC: Holm-Sidak $t = 3.22$, $p < 0.05$; NAc = $4.38 \pm 0.30$ synapses per μm, mPFC = $0.69 \pm 0.35$ synapses per μm). A representative image showing synapses formed between PV interneurons and a NAc-projecting pyramidal cell is shown in Fig. 3D.

To confirm the eGRASP results, immunogold labeling and electron microscopy were performed (Fig. 3E). Similar to the eGRASP results, we found a significant interaction (Fig. 3F; Shapiro–Wilk Normality Test $p = 0.08$; Two-way ANOVA $F_{(1,13)} = 5.04$, $p < 0.05$; $n = 4$ slices each from 4-5 mice) and a main effect of interneuron types ($F_{(1,13)} = 7.28$, $p < 0.05$). Thus, PV-positive interneurons form a similar number of synapses on NAc- and mPFC-projecting pyramidal cells (PV:NAc vs PV:mPFC: Holm-Sidak $t = 0.66$, $p > 0.05$; NAc = $0.14 \pm 0.02$ synapses per μm, mPFC = $0.16 \pm 0.04$ synapses per μm). SST-positive interneurons preferentially synapse on NAc-projecting pyramidal cells (PV:mPFC vs SST:mPFC: Holm-Sidak $t = 3.591$, $p < 0.05$; NAc = $0.13 \pm 0.01$ synapses per μm, mPFC = $0.05 \pm 0.01$ synapses per μm). A representative image is shown in Fig. 3G.

To further support the findings above, monosynaptic rabies virus tracing was used (Fig. 3H, I). Similar to the eGRASP and immuno-electron microscopy, we found a trend toward a significant interaction (Fig. 3J; Shapiro–Wilk Normality Test $p = 0.95$; Two-way ANOVA $F_{(1, 11)} = 4.138$, $p = 0.067$; $n = 2$-23 slices each from 3–4 mice). While PV-positive interneurons are synaptically connected to a similar number of pyramidal cells that project to the NAc and mPFC (NAc = $9.0 \pm 6.14\%$ of GFP+ cells, mPFC = $19.25 \pm 6.43\%$ of GFP+ cells), SST-positive interneurons seem to form fewer connections with mPFC-projecting pyramidal cells (NAc = $27.0 \pm 11.21\%$ of GFP+ cells, mPFC = $5.33 \pm 2.67\%$ of GFP+ cells). A representative image is shown in Fig. 3K.

## vHipp projections to the mPFC and NAc are differentially regulated by local interneurons

Finally, to determine if the observed anatomical differences are physiologically relevant, we used opto-electrophysiology to perform in vivo extracellular recordings from fluorescently labeled cells. To first confirm the validity of this technique, we expressed the red-shifted ChannelRhodopsin, C1V1, and YFP in vHipp pyramidal cells (Fig. 4A). Once a YFP-positive cell was identified, the micropositioner was used to move away from the cell to confirm a decrease in both action potential magnitude and emitted fluorescence as shown in Fig. 4B. Cells in which fluorescence increased by at least 130 percent over baseline were categorized as YFP-positive. YFP-positive cells showed a significant increase in fluorescence level compared to YFP-negative cells (Fig. 4C; Unpaired $t$-test $t = 6.09$, $p < 0.05$; YFP-negative cells = $100.8 \pm 2.49$ percent over baseline, YFP-negative cells = $153.8 \pm 11$ percent over baseline). Importantly, when the yellow laser was used to activate C1V1, only the YFP-positive cells increased their firing rate (Fig. 4D; Unpaired $t$-test $t = 3.77$, $p < 0.05$; YFP-negative cells = $85.88 \pm 8.60$ percent of baseline, YFP-positive cells = $141.7 \pm 12.82$ percent of baseline). Representative traces and YFP-positive cells are shown in Fig. 4E and F, respectively.

Next, we used opto-electrophysiology and optogenetics to record the activity NAc or mPFC-projecting neurons while inhibiting the activity of PV or SST interneurons (Fig. 4G)[42]. Differences in baseline firing rates were observed between the two pyramidal cell populations (Fig. 4H). Specifically, mPFC-projecting pyramidal cells appear to fire faster than pyramidal cells that project to the NAc (Unpaired $t$-test $t = 2.09$, $g < 0.05$; NAc-projecting cells = $0.56 \pm 0.11$ Hz, mPFC-projecting cells = $0.89 \pm 0.11$ Hz). When optogenetics was used to inhibit interneurons, we found that PV- and SST-positive interneurons differentially regulate pyramidal cells depending on their projection target (Fig. 4I, Supplementary Data 3; Shapiro–Wilk Normality Test $p < 0.05$; Kruskal–Wallis test, H $_{(4)} = 14.72$, $p < 0.05$). In line with the anatomical data, inhibition of PV-positive interneurons produces a significant increase in the firing rate of vHipp pyramidal cells that project to either the NAc and mPFC compared to controls that did not have HaloRhodopsin (Dunn's multiple comparisons test: Controls vs PV/NAc Z = 2.87, $p < 0.05$, Controls vs PV/mPFC Z = 2.83, $p < 0.05$; Controls = $103.4 \pm 4.99\%$ of baseline firing, PV/NAc = $168.3 \pm 22.33\%$ of

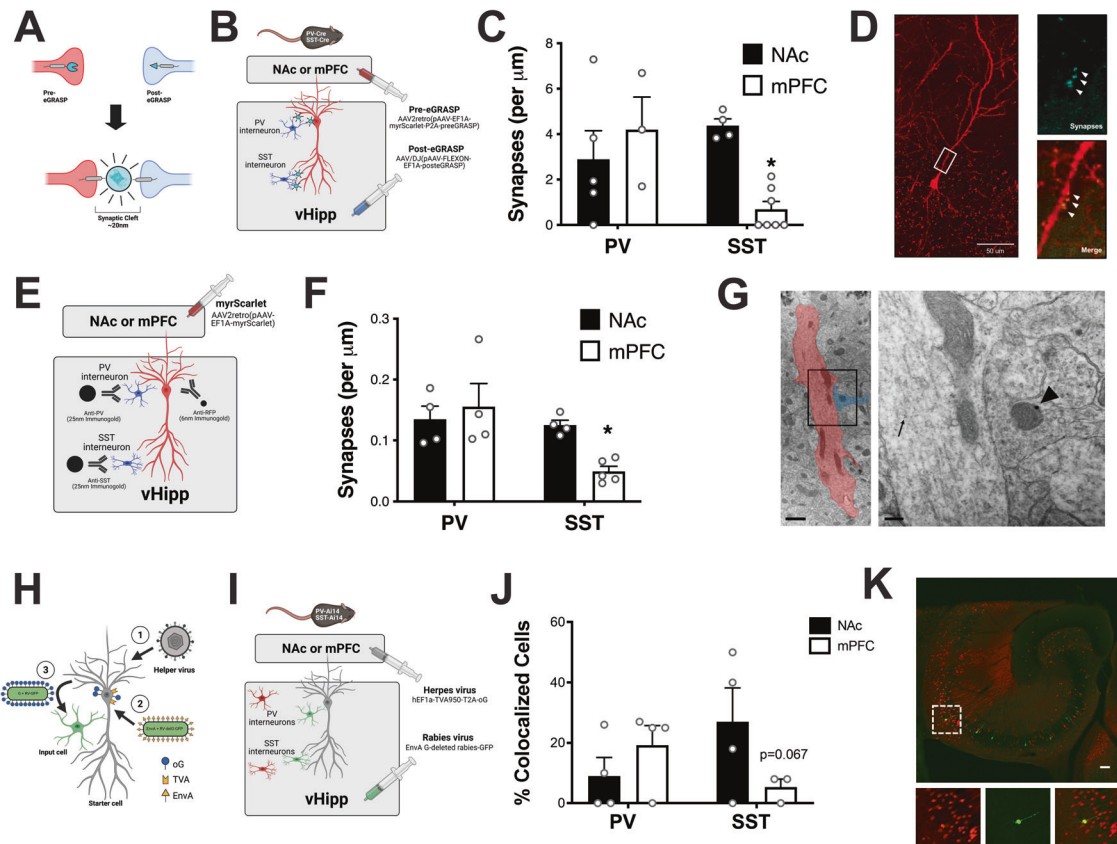

**Fig. 3 | Ventral Hippocampal (vHipp) projections to the medial prefrontal cortex (mPFC) and nucleus accumbens (NAc) are differentially innervated by local interneurons. A** Schematic demonstrating enhanced GFP Recombination Across Synaptic Partners (eGRASP) technology. **B** The pre-eGRASP component was expressed in mPFC- or NAc-projecting pyramidal cells and post-eGRASP component was expressed in vHipp parvalbumin (PV) or somatostatin (SST) cells. **C** PV- and SST-positive interneurons differentially connect to vHipp pyramidal cells depending on their projection target. Data analyzed by Two-way ANOVA, $p = 0.0126$. * denotes significant difference from NAc-projecting pyramidal cells: SST interneurons. $n = 3$ cells each from 3 (mPFC-projecting pyramidal cells: PV interneurons), 4 (NAc-projecting pyramidal cells: SST interneurons), 5 (NAc-projecting pyramidal cells: PV interneurons) or 7 (mPFC-projecting pyramidal cells: SST interneurons) mice per group. Representative image showing NAc-projecting pyramidal cell: PV interneuron synapses is shown in (**D**). Scale bar is 50 microns. **E** Immunogold labeling of pyramidal cell projections and specific interneuron subtypes. **F** PV- and SST-positive interneurons differentially connect to vHipp pyramidal cells depending on their projection target. Data analyzed by Two-way ANOVA, $p = 0.0429$. * denotes significant difference from NAc-projecting pyramidal

cells: SST interneurons. $n = 4$ dendrites each from 4 (NAc-projecting pyramidal cells: PV or SST interneurons; mPFC-projecting pyramidal cells: PV interneurons) or 5 (mPFC-projecting pyramidal cells: SST interneurons) mice per group. A representative image of mPFC-projecting pyramidal cell (red, arrows denote immunogold labeling of RFP): PV interneuron (blue, arrows denote immunogold labeling of PV) synapse is shown in (**G**). Scale bars are 1 μm and 200 nm. **H** Schematic of modified rabies virus. **I** The TVA receptor and optimized G protein was expressed in NAc- or mPFC-projecting pyramidal cells and the modified rabies virus was injected into the vHipp. **J** SST-positive interneurons show a trend toward less connectivity with vHipp pyramidal cells that project to the mPFC. $n = 2$-23 slices each from 3 (mPFC-projecting pyramidal cells: SST interneurons) or 4 (NAc-projecting pyramidal cells: PV or SST interneurons; mPFC-projecting pyramidal cells; PV interneurons) mice per group. Data analyzed by two-way ANOVA, $p = 0.0668$. A representative image of vHipp cells monosynaptically connected to NAc-projecting pyramidal cells (green) is shown in (**K**). SST interneurons shown in red. Scale bar is 100 microns. All data are presented as mean values +/- SEM. Source data are provided as a Source Data file.

baseline firing, PV/mPFC = 153.4 ± 16.12% of baseline firing). Inhibition of SST-positive interneurons also produced a significant increase in the firing rate of NAc-projecting pyramidal cells compared to controls (Dunn's multiple comparisons test: Z = 2.62, $p < 0.05$; SST/NAc = 149.3 ± 15.28% of baseline firing). Inhibition of SST-positive interneurons, however, had no effect on the firing rate of vHipp pyramidal cells that project to the mPFC (Dunn's multiple comparisons test: Z = 0.51, $p > 0.05$; SST/mPFC = 112.4 ± 10.21% of baseline firing). Representative electrophysiology traces before and during optogenetic inhibition of PV interneurons are shown in Fig. 4J. mCherry-positive PV interneurons are shown in Fig. 4K.

## Discussion

The hippocampus is a highly organized structure with dorsal and ventral regions (in humans, the posterior and anterior regions, respectively) comprising distinct structures, with unique input,

connectivity, gene expression, physiology, and behavioral regulation[1,22,23]. Recent evidence suggests that even within the vHipp, pyramidal cells are not as homogenous as once believed[35,36], rather, they have unique anatomy, molecular signatures, and behavioral effects. Indeed, we have previously demonstrated that vHipp projections to the medial prefrontal cortex (mPFC) and nucleus accumbens (NAc) differentially regulate schizophrenia-like behaviors[37]. Here we demonstrate that unique interneuron subtypes within the vHipp differentially regulate pyramidal cell populations based on their projection target.

Consistent with previous reports[32,43–45], vHipp pyramidal cells projecting to the NAc and mPFC comprise distinct, anatomically segregated cell populations with mPFC-projecting cells primarily located in the CA1 and NAc-projecting cells mainly in the subiculum. This is in line with work from others that used retrograde tracing to demonstrate that vHipp pyramidal cells that project to downstream brain

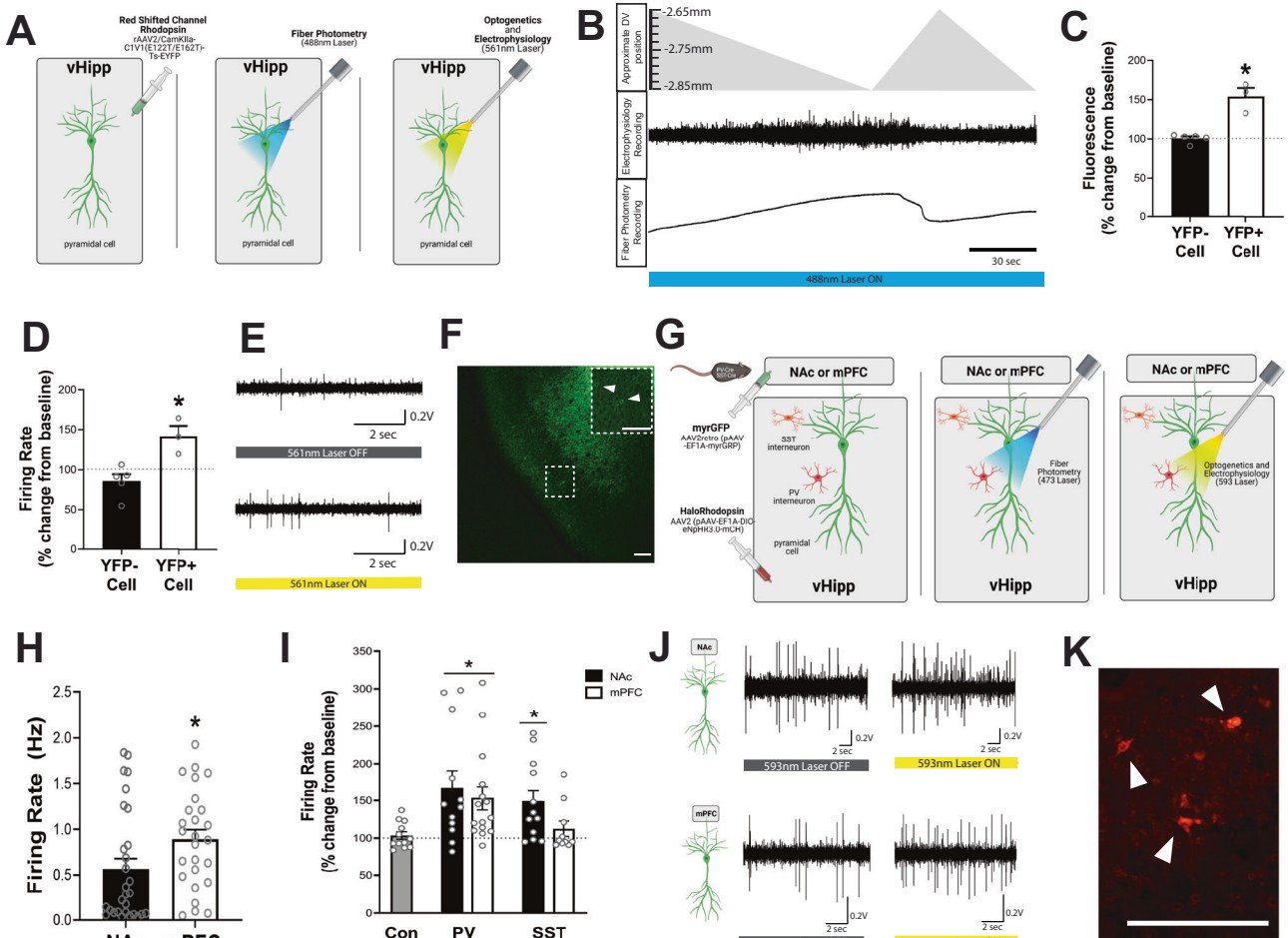

**Fig. 4 | Ventral Hippocampal (vHipp) projections to the medial prefrontal cortex (mPFC) and nucleus accumbens (NAc) are differentially regulated by local interneurons. A** The red-shifted ChannelRhodopsin, C1V1, and yellow fluorescent protein (YFP) were expressed in vHipp pyramidal cells and an opto-electric probe was used to record the electrical activity of fluorescently-labeled pyramidal cells before and during yellow laser activation. Emitted fluorescence and electrical activity were simultaneously recorded by the opto-electric probe as the micro-positioner was used to move through the vHipp as shown in (**B**). **C** An increase in fluorescence was observed in YFP-positive cells. $n = 3$ (YFP + ) or 5 (YFP-) cells per group. Data analyzed by unpaired $t$-Test, $p = 0.0009$. * denotes significant difference from YFP-negative cells. **D** A significant increase in firing rate was observed in YFP-positive cells. $n = 3$ (YFP + ) or 5 (YFP-) cells per group. Data analyzed by unpaired $t$-test, $p = 0.0093$. * denotes significant difference from YFP-negative cells. Representative traces are shown in (**E**). Representative image of YFP-positive cells are shown in (**F**). Scale bar is 100 microns. **G** myristoylated (myrGFP) was expressed in NAc- or mPFC-projecting pyramidal cells and halorhodopsin was expressed in

vHipp parvalbumin (PV) or somatostatin (SST) interneurons. An opto-electric probe shining a blue laser was used to identify GFP-positive pyramidal cells, then a yellow laser was used to activate halorhodopsin in interneuron subtypes. **H** Differential baseline firing rates were observed in vHipp pyramidal cells that project to the NAc and mPFC. Data analyzed by unpaired $t$-test, $p = 0.0418$. * denotes significant difference from NAc-projecting cells. $n = 26$ (mPFC-projecting) or 29 (NAc-projecting) cells per group. **I** PV- and SST-positive interneurons differentially regulate the activity of vHipp pyramidal cells depending on their projection target. Data analyzed by Kruskal–Wallis test, $p = 0.0053$. * denotes significant difference from control cells. Representative electrophysiological traces before and during PV interneuron inhibition are depicted in (**J**). Representative image of mCherry-positive PV interneurons in the hippocampus is shown in (**K**). Scale bar is 100 microns. $n = 10$ (mPFC projecting pyramidal cell: SST interneuron), 12 (NAc-projecting pyramidal cell: PV or SST interneuron, Control) or 15 (mPFC-projecting pyramidal cell: PV interneuron) cells from 5–6 mice per group. All data are presented as mean values +/- SEM. Source data are provided as a Source Data file.

regions involved in emotional processing, including the mPFC, NAc, lateral habenula, and basolateral amygdala are made up of anatomically segregated cell populations[35,36]. Further, the vast majority of vHipp pyramidal neurons project to only one brain region, with only a small percentage of cells sending efferents to multiple downstream targets[35,36]. It has been hypothesized that these dual-projecting cells may play a specialized role in regulating behavior.

In addition to the behavioral and anatomical differences in vHipp projections, we also demonstrate that mPFC- and NAc-projecting cells have unique transcriptional profiles. This is perhaps not surprising as comprehensive analyses of gene expression in the hippocampus have previously demonstrated differences in gene expression both across the dorsal-ventral axis[29–32] and between hippocampal subfields[30,31]. Importantly, using a previously collected dataset, (Cytosplore Viewer),

we determined that the differences in gene expression that we observed between mPFC-projecting and NAc-projecting cells could not be explained by anatomical location alone. Further, within individual hippocampal subfields, heterogenous cytoarchitecture and gene expression have suggested even further subdivisions[29,30,32]. In the ventral CA1, for example, clear laminar specificity is observed, with gene expression differences between not only the dorsal and ventral CA1 but also within superficial, middle, and deep pyramidal sublayers. Further, strong patterns of co-expression were observed in connected brain regions, including amygdala, lateral septum, bed nucleus of the stria terminalis, and hypothalamus, suggesting that discrete hippocampal projection pathways may display unique molecular signatures[29]. In line with the current findings, TRAP and RNASeq were recently used to show that vHipp projections, including those to the

mPFC and NAc, display unique transcriptional profiles. Interestingly, the mPFC-projecting cells had the highest number of differentially expressed genes compared to the other projections examined, with a strong enrichment of metabolic genes, including those involved in oxidative phosphorylation, and in genes associated with schizophrenia and neurodegeneration[35].

In the current experiments, gene expression analysis of mPFC- and NAc-projecting cells revealed differences in genes associated with synaptic plasticity, and RNAScope confirmed the differential expression of two specific genes in this pathway, *NEDD9* and *GRN*. *NEDD9* is a member of the CAS family of multidomain docking or scaffolding proteins that is generally down-regulated in adulthood[46]. During development, *NEDD9* regulates neuronal migration[47] and neurite outgrowth[48]. However, *NEDD9* remains highly expressed in the adult hippocampus and *NEDD9* knockout mice show spatial navigation deficits and reduced hippocampal spine density[49]. In humans, polymorphisms in the *NEDD9* gene have been associated with neurodegenerative diseases, including Alzheimer's and Parkinson's disease[50–54]. Similarly, polymorphisms in the *GRN* gene have also been associated with neurodegenerative diseases, including frontotemporal dementia[55,56]. The *GRN* gene encodes Progranulin, a pleiotropic protein involved in inflammation, development, and synaptic function. In mice, *GRN* knock-out has been shown to produce sex-specific deficits in anxiety-like behavior, motor coordination, and hippocampal synaptic plasticity[57]. Future experiments will explore the role of these proteins in regulating specific hippocampal circuits involved in cognition (i.e. mPFC-projecting neurons) in both health and disease.

In addition to the differences in anatomical localization and gene expression, the current studies also found a difference in the baseline activity of vHipp pyramidal cells that project to the mPFC versus NAc. Similar to patterns of gene expression, electrophysiological properties of cells also differ between hippocampal regions and along the dorsal-ventral axis. For example, slice electrophysiology demonstrated that ventral CA1 neurons are intrinsically more excitable than dorsal neurons, as a result of a more depolarized resting membrane potential and higher input resistance[26,28]. Further, firing patterns also differ between hippocampal subregions, with more bursting neurons in the subiculum than the CA1[58]. These differential firing patterns have also been shown to correspond to projection target, with neurons projecting to the NAc displaying a more regular firing pattern[59]. The increase in baseline firing rate that we observed in mPFC-projecting neurons may result either from intrinsic properties of the cell or extrinsic inputs to the cell.

Hippocampal pyramidal cells receive input from ~35 other brain regions, including the medial preoptic area, posterior amygdala, nucleus reuniens, and paraventricular nucleus of the thalamus, and it has been shown that these inputs are dependent on both the spatial location of cells and their projection target[36]. However, the vast majority of inputs to vHipp cells come from local sources[36]. Dramatic differences between local inputs to mPFC- and NAc- projecting neurons have been suggested using a cell transplantation approach to examine the role of discrete interneuron subtypes in symptoms associated with schizophrenia. Specifically, we previously demonstrated that transplantation of somatostatin interneurons (SST) into the vHipp of a rodent model, rescue deficits associated with negative and cognitive symptoms while transplantation of parvalbumin (PV) interneurons reverses deficits across all symptom domains[19]. This provided the foundation for our hypothesis that PV and SST interneurons may differentially innervate hippocampal pyramidal neurons based on their projection target. Here, we demonstrate that PV and SST interneurons differentially target pyramidal cell projections, with PV interneurons forming a similar number of synapses on pyramidal cells that project to both the NAc and the mPFC while SST interneurons primarily target vHipp pyramidal cells projecting to the NAc.

Importantly, this observation was made in separate experiments using distinct anatomical techniques (i.e. eGRASP, immunogold labeling and electron microscopy). Monosynaptic rabies tracing showed a similar trend; however, the inability to identify starter cells limits the interpretation of this study as the location of the injection site and rate of viral transfection can affect the total number of labeled cells. Moreover, these anatomical observations are physiologically relevant as optogenetic silencing of PV interneurons increased the activity of both mPFC- and NAc- projecting neurons whereas SST interneuron inactivation only altered the activity of NAc-projecting pyramidal cells. These findings are in line with our previously collected behavioral data demonstrating that PV interneuron transplants improve behaviors associated with both the vHipp-NAc and the vHipp-mPFC pathway while SST interneuron transplants only attenuate behavioral deficits associated with the vHipp-NAc pathway, but not those associated with vHipp-mPFC pathway.

The current experiments demonstrate differential synaptic connectivity and functional regulation of vHipp projection neurons by specific interneuron subpopulations. While some have suggested that inhibitory interneurons provide "blanket inhibition" by non-selectively targeting cells within close proximity[60,61], more recent evidence suggests that interneuron subtypes may also provide more targeted regulation of local cells[62]. For example, specific interneuron subtypes that preferentially target other interneurons have been identified in the hippocampus[63,64]. Further, specific interneuron subtypes within the hippocampus may also differentially target pyramidal cell populations that project to certain brain regions. In the medial entorhinal cortex, cholecystokinin and cannabinoid type 1 receptor (CB1R)-expressing basket cells preferentially target pyramidal cells that project outside of the hippocampus[65]. In addition, it has recently been shown that PV-positive interneurons preferentially innervate hippocampal pyramidal cells that project to the amygdala[66]. Together, these results suggest that vHipp pyramidal cell populations can not only be segregated by anatomical location, projection target, and transcriptional profile, but also display distinct connectivity within the hippocampus. Understanding these unique vHipp cell populations may help identify novel targets for the treatment of specific symptom domains of schizophrenia and other psychiatric disorders that involve hippocampal dysfunction.

## Methods

### Mice

Male and female mice were purchased from Jackson Laboratories (C57BL/6, PV-Cre strain #017320, SST-Cre strain #013044, Ai9#007909) to use as breeders. Male and female pups were weaned at postnatal day 21, and all experiments included male and female adult animals (12–24 weeks old). Mice were maintained on a 12 h/12 h light/dark cycle with food and water available *ad libitum*. The animal housing rooms were maintained at 75 ± 3 degrees Fahrenheit and ~55–58 percent humidity. All experiments were performed in accordance with the guidelines outlined in the USPH Guide for the Care and Use of Laboratory Animals and were approved by the Institutional Animal Care and Use Committee at either the University of Texas Health Science Center at San Antonio or the University of Texas at Austin.

### Stereotaxic surgeries

To inject viral constructs, mice were anesthetized using Fluriso (1–4% Isoflurane, USP with oxygen flow at 1 L/min) and placed in a stereotaxic apparatus. Guide cannula (22 Gauge, Plastics One) were aimed at the vHipp (A/P-2.9, M/L ± 2.8, D/V-3.0), mPFC (A/P + 1.8, M/L ± 0.3, D/V-2.0), or NAc (A/P + 1.2, M/L ± 1.0, D/V-3.5)[67]. The virus was injected manually at a rate of ~0.5 µl/20 s. through an injector (28 Gauge, Plastics One), which extended 1 mm past the cannula tip. The injector was left in place for 3 min after the injection. Mice received one dose of

peri-operative analgesia (ketaprofen, 5 mg/kg, i.p.) and were allowed 6 weeks to recover before electrophysiology recordings or tissue collection.

## Projection tracing

To retrogradely label vHipp projections, AAV2retro-myrGFP (pAA-V[Exp]-EF1A > {myrGRFP}:WPRE, $1 \times 10^{13}$ GC/ml, Vector Builder) or AAV2retro-myrScarlet (pAAV[Exp]-EF1A > {myrmScarlet-1}:{P2A}:{yellow pre-eGRASP(p32)}:WPRE, $4.29 \times 10^{13}$ GC/ml, Vector Builder) were injected into the mPFC or NAc of wild-type mice (Fig. 1A). After 6 weeks, animals were anesthetized using Fluriso (1–4% Isoflurane, USP with oxygen flow at 1 L/min), then transcardially perfused with saline followed by 4% formaldehyde. The brains were removed and a coronal slice was made through the brain at the level of the optic chiasm. Coronal sections were made through the anterior portion of the brain to confirm injection size and location. From the posterior portion of the brain, 50 µm horizontal sections were cut through the vHipp using a cryostat. Sections were mounted on slides, and coverslipped using prolong gold antifade reagent, then imaged on a Zeiss LSM710 confocal microscope. The number of GFP- or Scarlet-positive neurons was determined on 2–4 sections per animal from 4 animals.

## RNA sequencing

RNASeq was used to determine the transcriptional profile of vHipp projection neurons (Fig. 2A). Briefly, a retrograde virus (pAA-V[Exp]-EF1A > {myrmScarlet-1}:{P2A}:{yellow pre-eGRASP(p32)}:WPRE, $4.29 \times 10^{13}$ GC/ml, Vector Builder) was injected into the mPFC or NAc to express myrScarlet in vHipp projection neurons of wild-type animals. After 6 weeks, mice were anesthetized using Fluriso (1–4% Isoflurane, USP with oxygen flow at 1 L/min) and the brain was removed. Then, vHipp tissue was dissected and minced, placed in Hibernate A (Gibco), and spun for 2 min at 425 $x$ $g$. Supernatant was replaced with accutase (Thermo Fisher) and tubes were placed on a shaking platform at 4 °C for 30 min. Tissue homogenates were spun for 2 min at 425 $x$ $g$ and the pellet was resuspended in Hibernate A. Cells were dissociated by trituration with fire polished glass pipettes. Supernatant containing cells were combined and filtered through a 75 µm strainer, then stored on ice. Cells were sorted using a BD FACSAria III (Franklin Lakes, NJ, USA, 70 µm nozzle, 50 psi) and analyzed using FACS Diva 6.1.3 software (BD). After excluding doublets, Scarlet-positive cells were collected in RNA-Later (Invitrogen). Gating strategies are shown in Supplementary Data 4. RNA was prepared using Qiagen RNEasy Micro kit and sent to LC Biosciences for low input RNA sequencing. Briefly, a Poly (A) RNA sequencing library was prepared using Illumina's TruSeq-stranded-mRNA preparation protocol. RNA integrity was determined using Agilent Technologies 2100 bioanalyzer. Poly (A) tail-containing mRNAs were purified using oligo-(dT) magnetic beads with two rounds of purification, followed by fragmentation using divalent cation buffer in elevated temperature. After DNA library construction, quality control analysis was performed using Agilent Technologies 2100 Bioanalyzer High Sensitivity DNA chip. Pair-ended sequencing was performed on Illumina's NovaSeq 6000 sequencing system. To analyze the data, Cutadapt and perl scripts were used to remove reads containing adapter contamination, low quality bases, or undetermined bases. Then sequence quality was verified using FastQC. HISAT2 was used to map reads to the mouse genome and mapped reads were assembled using StringTie. Next, all transcriptomes were merged to reconstruct a comprehensive transcriptome using perl scripts and gffcompare. After the final transcriptome was generated, StringTie and edgeR was used to estimate the expression levels of all the transcripts. StringTie was used to perform expression level for mRNAs by calculating FPKM. The differentially expressed mRNAs were selected with log2 (fold change)>1 or log2 (fold change) <−1 and with statistical significance ($p < 0.05$) by R package edgeR.

## RNA Scope

RNAScope was used to confirm differential gene expression of *NEDD9* and *GRN*, two genes identified by RNASeq as being enriched in mPFC projecting neurons (Fig. 2D). Fluorogold (Fluorochrome, 0.3 µl, 2%) was injected into either the mPFC or NAc to label vHipp projection neurons in wild-type mice. After 1 week, mice were anesthetized using Fluriso (1–4% Isoflurane, USP with oxygen flow at 1 L/min), then transcardially perfused with saline, followed by 4% formaldehyde. The brain was removed and cryoprotected in increasing concentrations (10%, 20%, 30%) of sucrose before freezing. Twenty µm horizontal sections were cut through the vHipp and collected on slides, which were dried for 1 hr at −20 °C, then stored at −80 °C until testing. Slides were fixed in 4% paraformaldehyde for 15 min at 4 °C. RNAScope was performed according to manufacturer's instructions (ACD Bio). Briefly, tissue was dehydrated in increasing concentrations of EtOH, incubated in protease solution for 30 min, then washed. *NEDD9* and *GRN* probes were hybridized at 40 °C for 2 hr before washing. Four amplification steps were performed before slides were coverslipped with prolong gold. Slides were imaged on a Zeiss LSM710 confocal microscope and IMARIS image analysis software was used to determine the number of GFP-positive puncta per fluorogold-labeled cell. At least 3 cells were analyzed per animal from 4-6 mice per group by an experimenter blind to the experimental condition.

## eGRASP

Detection of synapses between genetically defined cell populations was performed using the enhanced GFP recombination across synaptic partners (eGRASP) technique, which requires the complementation of two non-fluorescent GFP fragments (Fig. 3A)[68]. To specifically identify synaptic connections between vHipp projection neurons and local interneurons using eGRASP, a retrograde virus (AAV2retro) expressing myrScarlet (pAAV[Exp]-EF1A > {myrmScarlet-1}:{P2A}:cyan pre-eGRASP(p32)}:WPRE, $9.02 \times 10^{13}$ GC/ml, Vector Builder) was injected into either the mPFC or NAc of PV- or SST-Cre mice. Concomitantly, an AAV-DJ/8 containing the floxed post-eGRASP component (pAAV[-FLEXon]-EF1A > LL:rev({post-eGRASP}):rev(LL):WPRE, $2.63 \times 10^{13}$ GC/ml, Vector Builder) was injected into the vHipp. Six weeks later, mice were anesthetized using Fluriso (1–4% Isoflurane, USP with oxygen flow at 1 L/min), then transcardially perfused with saline, followed by 4% formaldehyde. The brain was removed and 200 µm sections were cut through the vHipp using a vibratome (Leica). Sections were mounted on slides and coverslipped using prolong gold antifade reagent. A Zeiss LSM710 Confocal microscope was used to identify and image myrScarlet-positive pyramidal cells in 3 dimensions. IMARIS image analysis software was used to reconstruct the myrScarlet-positive cells and to identify GFP-positive synapses within 3 µm of the pyramidal cell. At least 3 cells per animal from 3-7 mice per group were analyzed by an experimenter blind to the experimental condition.

## Immunogold labeling and electron microscopy

Electron microscopy and immunogold labeling were used to confirm the eGRASP data in wild-type mice (Fig. 3E). To identify synaptic connections between vHipp projection neurons and local interneurons, a retrograde virus expressing myrScarlet label (pAAV[Exp]-EF1A > {myrmScarlet-1}:{P2A}:yellow pre-eGRASP(p32)}:WPRE, $4.29 \times 10^{13}$ GC/ml, Vector Builder) was injected into either the mPFC or NAc. Six weeks later, mice were anesthetized with Fluriso (1–4% Isoflurane, USP with oxygen flow at 1 L/min), then perfused with saline, followed by 4% paraformaldehyde/1% glutaraldehyde. The mouse brain was removed and the vHipp was dissected and stored in fixative. Ultrathin (90 nm thick) sections were prepared and immunohistochemistry was used to label RFP-positive pyramidal cells and PV- or SST-positive interneurons. Briefly, grids were washed, then incubated in blocking buffer (5%BSA/5%NGS) followed by primary antibodies for RFP (Chicken anti-

RFP, Synaptic Systems Catalog #409006, 1:50) and PV (Rabbit anti-PV, Abcam Catalog #11427, 1:50) or SST (Rabbit anti-SST, Abcam Catalog #183855, 1:50). After washing, grids were then incubated in secondary antibodies (Goat anti-chicken 6 nm immunogold, EMS Diasum Catalog #25587, 1:25; Goat anti-rabbit 25 nm immunogold, EMS Diasum Catalog #25116, 1:25). Uranyl Acetate (2%) was used to increase contrast staining. Grids were imaged using a JEOL 1400 Transmission electron microscope. RFP-positive dendrites adjacent to PV- or SST-labeled interneurons were identified, and the length of the dendrite and the number of synapses formed between the two cells was determined. Only synapses containing synaptic vesicles, mitochondria, and a clear synaptic cleft were included in the analysis. At least 4 sections per animal from 4–5 mice per group were analyzed by an experimenter blind to the experimental condition.

### Monosynaptic rabies virus tracing

To further confirm the pattern of connectivity between hippocampal projection neurons and specific interneuron subtypes, a modified rabies virus was used in PV-Ai14 or SST-Ai14 mice (Fig. 3H, I)[69,70]. Briefly, herpes virus (hEF1a-TVA950-T2A-oG, $2.5 \times 10^9$ GC/ml, Mass General Hospital Vector Core) was injected into the NAc or mPFC to express the avian TVA950 receptor, required for the EnvA-coated rabies virus to access the cell, and optimized rabies G protein, an envelope protein required for transneuronal transfer of rabies virus, in pyramidal cells that project to these regions (starter cell). Four weeks later, rabies virus (EnvA + RV-delG-GFP, $5.0 \times 10^7$ GC/ml, Salk Institute), modified by the addition of an envelope protein from avian ASLV type A, deletion of rabies G protein, and addition of GFP to the viral genome, was injected into the vHipp in order to infect starter cells expressing the TVA receptor. Importantly, the G expressed in trans results in rabies virus particles that express the G in their viral envelope and can bud out from the starter cells and infect synaptically connected neurons (input cells). One week later, mice were anesthetized with Fluriso (1–4% Isoflurane, USP with oxygen flow at 1 L/min), then transcardially perfused with saline, followed by 4% formaldehyde. The brain was removed and 50 μm sections were cut through the vHipp using a vibratome (Leica). Sections were mounted on slides and coverslipped using prolong glass antifade reagent. Slides were imaged on a Nikon Ni-E Upright Motorized fluorescent microscope and the number of GFP-positive (monosynaptically connected to pyramidal cell projections), tdTomato-positive (specific interneuron subtypes), and colocalized cells we quantified. Up to 23 sections per animal from 3-4 mice per group were analyzed by an experimenter blind to the experimental condition.

### Opto-electrophysiologic recordings

To perform in vivo extracellular recordings from fluorescently labeled cells, opto-electrophysiology was used as previously described in[42]. To first confirm the validity of this technique, an AAV2 (rAAV2/CamKIIa-C1V1(E122T/E162T)-Ts-EYFP, $4.9 \times 10^{12}$ GC/ml, UNC Gene Therapy Center Vector Core) was used to express the red-shifted Channel Rhodopsin, C1V1, in vHipp pyramidal cells (Fig. 4A). After a 4-week recovery period, animals were anesthetized with Fluriso (1–4% Isoflurane, USP with oxygen flow at 1 L/min) and placed in a stereotaxic apparatus. To identify fluorescently labeled cells while simultaneously recording their electrical activity, an opto-electric probe (Doric, SCRT_10) was made of a dual core optical fiber, with a 500 μm optical core and a 250 μm hollow core for electrolyte filling, and pulled to a 10 μm diameter tip at one end. The probe was connected to a 5 ports fluorescence mini cube (Doric, FMC5) and lowered into the vHipp using a hydraulic microdrive (Narishige, Model MO-10). A 50-mW 488 nm laser (Doric) was used to pass blue light through an optical fiber and into the optical core of the opto-electric probe, in order to excite the YFP fluorophore. Emitted fluorescence was collected by the opto-electric probe and passed in the opposite direction through the same optical fiber to a photomultiplier tube (PMT, Hamamatsu Model C10709). The PMT output was

converted to a digital signal using AD Powerlab and analysis was performed using LabChart Software (AD Instruments). Once a YFP-positive cell was identified, the micropositioner was used to move away from the cell to confirm a decrease in both firing rate and emitted fluorescence. After moving back to the cell, the 488 nm laser was turned off and the hollow core of the opto-electric probe, which was filled with 2 M NaCl and connected to a traditional headstage and electrophysiology recording system, was used to record baseline firing rate for at least 5 min. Previous studies have confirmed that the optical and electrical signals originate from the same cell by demonstrating that the axial position of the two signal peaks are less than the diameter of the neuron[42]. Then, a 50 mW 561 nm laser (Coherent OBIS Laser Systems) was used to pass yellow light through the opto-electric probe to activate C1V1 while the firing rate was recorded for an additional 5 min. Upon the completion of recordings, animals were rapidly decapitated and the brain was removed to confirm fluorescence. The percent change in fluorescence and the percent change in firing rate from baseline were analyzed using LabChart Software. Data was analyzed from at 3-5 cells per group from 2 mice.

Next, opto-electrophysiology was used to determine the functional regulation of vHipp pyramidal cells by PV- or SST-positive interneurons (Fig. 4G). Briefly, AAV2retro-myrGFP (pAAV[Exp]-EF1A > {myrGRFP}:WPRE, $1 \times 10^{13}$ GC/ml, Vector Builder) was injected into either the mPFC or NAc to label vHipp projection neurons in PV-Cre or SST-Cre mice. At the same time, a virus expressing a floxed version of halorhodopsin (rAAV2/EF1a-DIO-eNpHR3.0-mCherry-WPRE, $5.1 \times 10^{12}$ GC/ml, UNC Gene Therapy Center Vector Core) was injected into the vHipp. Control animals received the AAV2 retro virus to label vHipp projection neurons but no halorhodopsin. After a 6-week recovery period, opto-electrophysiology was performed essentially as described above. Briefly, a 100-mW 473 nm DPSS laser (OEM Laser Systems) was used to pass blue light into the optical core of the opto-electric probe, in order to excite the GFP fluorophore. Once a GFP-positive cell was identified, the 473 nm laser was turned off and the baseline firing rate was recorded for at least 5 min. Then, a 100 mW 593 nm DPSS laser (OEM Laser Systems) was used to pass yellow light through the opto-electric probe to activate HaloRhodopsin while firing rate was recorded for an additional 5 min. Upon the completion of recordings, animals were rapidly decapitated and the brain was removed to confirm fluorescence. Baseline firing rate and the percent change from baseline were analyzed using LabChart Software. Only pyramidal cells displaying GFP fluorescence and a firing rate <2 Hz (Ranck Exp Neurol 1973) were included in the analysis. Cells that experienced >20 percent decrease in firing rate in the presence of the 593 nm laser were excluded. Data was analyzed from at least 10 cells from 5–6 mice per group.

### Data analysis

RNAScope, eGRASP, electron microscopy, and rabies data were analyzed using a two-way ANOVA (projection target x interneuron subtype). When significant main effects were detected, the Holm-Sidak test was used for post-hoc comparisons. Baseline firing rates were analyzed using an unpaired *t*-test. Firing rate changes from baseline were analyzed using the Kruskal–Wallis test and Dunn's multiple comparison post-hoc analysis. All data are represented as the mean +/- SEM, unless otherwise stated, with n values representing the number of mice per group. Significance was determined at $p < 0.05$. All data were analyzed and graphed using SPSS (IBM) or Prism version 9 (GraphPad Software Inc.).

## Data availability

The RNASequencing Data is freely available online at the GEO (Gene Expression Omnibus) public functional genomics data repository (accession number GSE244159). Source data are provided with this paper. Raw data will be available via email upon request. Source data are provided with this paper.

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

## Acknowledgements

This work was supported by R00-MH121355 to J.J.D, a Young Investigator Award supported by The Pfeil Foundation, Inc. and the Brain & Behavior Research Foundation to J.J.D., Merit Awards #BX004693 and #BX004646 from the United States Department of Veterans Affairs, Biomedical Laboratory Research and Development Service and National Institutes of Health grants R01-MH090067 to D.J.L, and T32-NS082145 and F31-MH127890 to H.B.R. Cell sorting was performed by the Flow Cytometry Shared Resource Facility, which is supported by UT Health San Antonio, NIH-NCI P30 CA054174-20 (CTRC at UT Health) and UL1 TR001120 (CTSA grant). Confocal images were generated in the Core Optical Imaging Facility which is supported by UT Health San Antonio and NIH-NCI P30 CA54174. Electron microscopy images were generated using the Electron Microscopy Core at UT Health San Antonio. Light microscopy was performed at the Center for Biomedical Research Support Microscopy and Imaging Facility at UT Austin (RRID# SCR_021756). Figures 1a, 1c, 2a, 2d, 3a, 3b, 3e, 3h, 3i, 4a, 4g, and 4j were created with BioRender.com.

## Author contributions

D.J.L and J.J.D.—conceptualized the studies; D.J.L, H.B.R., A.M.B. and J.J.D.—performed the experiments; J.J.D. and H.B.R.—analyzed the data; J.D.—wrote the manuscript; D.J.L, H.B.R., A.M.B. and J.J.D.—edited the manuscript.

## Competing interests

The authors declare no competing interests.
