## [Peer Review File · Nature Communications]

Discrete hippocampal projections are differentially regulated by parvalbumin and somatostatin interneuronsREVIEWER COMMENTS

Reviewer #1 (Remarks to the Author):

The authors present some important work showing that projections from the ventral hippocampus (vHC) to the prefrontal cortex (mPFC) vs. to the nucleus accumbens (NAc) originate from distinct projection neurons spread across vCA1 and vSub; such distinct populations also show some differences in gene expression and - importantly - are differentially inhibited by SST-interneurons, while PV-interneurons innervate them both to roughly equal extent (although the variability within each population is extremely high, possibly explained by other factors not further analysed). The manuscript is very well written, the topic is very timely and the applied and combined methods are impressive and very state-of-the art. I have a few concerns, as follows:

Major points

A. Fig. 1-2: major part of the discovered projection-specificity of certain vHC pyramidal neurons (PNs) seems to be mediated simply by a distinction between vCA1 and vSub according to Fig. 1C - and it was known before that the vHC targets the mPFC mainly from CA1 (Jay TM, Witter MP (1991), *J Comp Neurol* 313: 574-586) and the NAc mainly from the ventral subiculum (see e.g. studies from A. Grace and others), see also Bienkowski et al., *Nat Neurosci*, 2018 for a more recent and more detailed description; consequently, also the differential gene expression may be a majorly explained by differences between vCA1 and vSub PNs; is it possible to address this hypothesis with the data of the others and other existing gene expression datasets from vCA1 and vSub (e.g. from the Allen Institute, accessible through CytosploreViewer, or from the lab of Sascha Nelson)?

B. Fig. 4E / statistics: the analysis with a Kruskal-Wallis one-way ANOVA is unclear here; given that a parametric ANOVA was used with the rather non-parametric small-N distribution in 3C/F in order to be able to report interactions, it is unclear why with the more "parametric-looking"/high-N dataset in 4E no such ANOVA is used. Please calculate a univariate ANOVA (excluding controls) with the factors interneuron-type and projection-target to calculate the interaction between those two in order to be able to argue for a target-specificity of interneuronal connectivity; also, for statistical analysis of ratios, prior log-transformation should be considered; the one-way ANOVA is not wrong and can be done in addition, but given that the key claim of this dataset and the paper overall seems to be the target-specificity of interneuronal connectivity, one would really want to see a significant interaction to support this claim; if the claim is somewhat toned down to a differential connectivity of SST-interneurons as such (i.e. disregarding the connectivity-difference between PV and SST at this stage), then such a 2-way ANOVA and sign. interaction would not be necessary, but then one could even analyse PV and SST separately or only including controls in each analysis

C. Fig. 4: a lot of the conclusion of this paper is grounded on the trust in the method of fluorescently guided in vivo recording of individual neurons used here, which in turn relies - somewhat blindly - on the suitability of the opto-electric probe to always record electrophysiological signals from the same neuron from which it also captures a fluorescence signal; is there any validation data (e.g. citable papers on this method) to support this experimental assumption (and possibly reports some error rates)? What about fluorescence light from nearby neurites of a recorded cell?

Minor points

1. the authors write the Introduction as if a hyperactive vHC or vHCPFC connection would be clearly detrimental in the negative symptom domain, but neither integrate nor cite their own study demonstrating that the activation of this region/pathway exerts an anti-depressant response which even appears to be part of the long-term anti-depressant effect of ketamine (Carreno et al., *Mol*

Psych, 2015); can the authors offer a more comprehensive view on the role of hyperactivity of this region/pathway for the negative symptom domain?

2. p.5, please add details on the surgery, such as the analgesic regime used peri-operatively (aside from isoflurane anaesthesia), infusion speed and volumes, needle size, type and size of implanted cannula; also please detail further the opto-electrophysiological system used for data in Fig. 4 (e.g. individual components, order numbers, etc.)

3.p. 5 and later; please unify the writing of the greek "micro" in μm

4. Fig. 2B, the gene names are not legible, please consider re-arranging Fig. 2 and enlarge the panel B; also the relevance of the identified genes presented here is not clear relative to the context of the physiological and behavioural differences of the two analysed projections - can anything more functionally relevant be drawn from this dataset, except for showing that the populations are different (e.g. two GPRs come up, but with uncertain neural relevance, another GPCR might be relevant, ...)?

5. Fig. 3 C, F, J ... the statistics displayed in the graph is confusing; in C and F one assumes that the asterisk indicates the post-hoc test within the SST-sub-groups, but in J the $p = 0.067$ seems to be what the authors cite as p-value of the interaction in the overall ANOVA; please indicate and/or state clearly the p-values of the subtype-target interaction, and of the pairwise post-hoc tests comparing target within SST and comparing subtype within mPFC-target (if relevant, seems to be different); also, given the small n-number (= mice, according to figure legend) and high variability in these essential panels, please state how many slices were analysed per mouse; apart from that, is there an explanation why the scale of the y-axes in C and F is so different, even though seemingly the same parameter is displayed? Final remark: the n-numbers are - understandably, given the used methods - rather small in these three panels which could be perceived as unconvincing, but the authors may want to strengthen the notion the three datasets clearly support each other, and are hence quite convincing as a set despite small n (just a suggestion)

6. Fig. 4C, it is unclear what is meant to be shown here and why a sudden (!) strong step in calcium influx occurs after ca. 2s of excitation - is the timeline the approach to the cell?; please describe more in detail, and possibly show more traces to illustrate the process

7. Fig. 4F, the text says traces "before and after optogenetic inhibition", but the figures seems to show traces before and during inhibition - which is correct? Which interneuron-type is inhibited here (and shown in 4G)?

8. maybe the title is slightly misleading in that it sounds too general ("distinct microcircuits regulate") relative to the underlying data; the key point really is that NAc-projecting neurons are distinct from mPFC-projecting neurons (Fig. 1-2) and that only the former get strongly inhibited by SST-interneurons (Fig. 3-4) - I am not requesting any change to the title, but just strongly suggest re-consider exchanging it for a more informative and accurate one; note that some of this over-generalization also appears in Abstract and Discussion, where it could be re-placed with more concrete statements

Reviewer #2 (Remarks to the Author):

The manuscript describes a study that seeks to establish the structural and functional relationship between PV and SST interneurons and distinct populations of ventral hippocampal projection neurons (NAc-projecting vs mPFC-projecting). The author's previous work has shown that transplantation of PV and SST into the ventral hippocampus may be a promising treatment for schizophrenia symptoms and the differential effects of PV vs. SST transplantation led to the main hypothesis that PV and SST interneurons are connected to distinct projection neuron subpopulations. The authors use a variety of

viral tracing and physiological techniques to establish this differential connectivity. Although the manuscript is well-written, I have concerns about the quantification and analysis of the data and how that influences the interpretation of the results.

Major Concerns:

- 1) This study often discusses the ventral hippocampus (vHipp) as one solitary structure and the authors proceed to dissect the NAC-projecting vs. mPFC-projecting population. However, as the first dataset shows, the mPFC-projecting population is largely distributed within the CA1 whereas the NAC-projecting population is primarily in the ventral SUB. Because of this segregated distribution, it is unclear if the differences shown in the other experimental data are actually projection-specific (mPFC-projecting vs. NAc-projecting) or an effect of anatomical location (CA1 neurons vs SUB neurons). If the authors analyze CA1 and SUB groups within their data they could address this and strengthen their overall argument that these effects are projection-specific (ex. compare mPFC-projecting CA1 neurons vs mPFC-projecting SUB neurons). The CA1 and SUB are functionally distinct brain regions that should be treated independently.
- 2) The methodology described for the quantification of the experimental datasets is unclear. First, some characterization of the injection set size and location is critical for the interpretation of the results as this can affect the distribution of the retrogradely-labeled cells within the sampling area. As for the sampling area, it's unclear where the dorsoventral location of the horizontal section and if this is in fact sampling from the area with the most labeled cells (this should depend on the injection site locations). Overall, the sparse number of reportedly labeled cells (~5-6 cells per slice) suggests either the sampling area is insufficient or the AAV2retro viruses are not labeling the full population. Viral tropism with AAVretro viruses is an additional concern here as these viruses are known to weakly infect some cell types (ex. layer 6 cortical pyramidal cells). Non-viral tracers such as Fluorogold or Cholera toxin would be an alternative.
- 3) The RNAscope methodology states that the quantification was used to "determine the number of GFP-positive puncta per fluorogold-labeled cell" but the results are reported as "puncta per μm " which is not appropriate to truly determine if cells are differentially expressed. It is not clear how meaningful this effect is and how it compares to the differential expression demonstrated by the flow cytometry experiments. In summary, RNAscope data should be presented as number of puncta per cell and a comparison to flow cytometry would help to strengthen to two differential gene expression analyses.
- 4) The eGRASP experiments also present the data as "synapses per μm " when this data should be presented as "synapses per cell" if the goal is to understand how SST and PV interneurons are differentially connected to different projection populations. Presenting the data like this would lead to overrepresentation of highly connected cells. Furthermore, the Two-way ANOVA does not say the effect is significant and the p value is not listed. P value and significance should be included and if the effect is not significant than it is not appropriate to perform post-hoc tests.
- 5) The monosynaptic rabies tracing experiment should be analyzed by quantifying the number of SST and PV GFP positive cells as a ratio to the number of labeled "starter cells" within individual animals as the number of starter cells can vary between animals. The quantification as it currently is describes how PV and SST represent the percentage of total monosynaptic input but don't help explain how many interneurons synapse on to an individual projection neuron cell type. Furthermore, judging by the graph in figure 3J, it appears that 5 of the 20 animals in these experiments didn't have any monosynaptically connected interneurons labeled (0%).

Minor Concerns

- 1) The methods sections states that quantification was performed on "2-4 sections per animal" but the number of animals is not listed in methods (only in figure caption). All of the sample size (n) should be reported in the methods section.
- 2) Figure 1, the fluorescent images are too small to discern the detailed labeling. I would suggest enlarging these images and minimizing the schematic.
- 3) The 4% fluorogold concentration used in RNAscope experiments is likely too high. Greater than 2% is considered cytotoxic and likely caused necrosis at the injection site. This could effect the labeling of projection neuron populations in the vHipp.

- 4) Why are the number of synapses/um reported by the eGRASP experiments approximately 20-30 times higher than the synapses/um reported by the EM experiments?
- 5) Figure 3D shows a representative eGRASP image but it is unclear from which experiment this image is from. Do PV and SST neurons synapse on to similar dendritic compartments for mPFC- and SST-projecting populations? In addition, Imaris could be used to reconstruct the 3D morphology of the pyramidal cells and map the synaptic contacts to the dendrites.
- 6) Stereotactic coordinates are given for cannula placement but it is unclear which stereotactic atlas was used for targeting. Are vHipp injections more in the CA1, SUB, or in between both?

Reviewer #3 (Remarks to the Author):

This manuscript explores an intriguing hypothesis that PV and SST interneurons in vHipp differentially innervate and regulate mPFC and NAc projecting neurons. This hypothesis is inspired by a finding that transplanting SST and PV neurons to vHipp differentially regulates behaviors associated with mPFC and NAc projections. Data from a variety of retrograde viral tracing strategies and in vivo electrophysiology with targeted optogenetic silencing of PV or SST interneurons demonstrate that, while PV interneurons innervate both mPFC and NAc projecting vHipp neurons, SST interneurons preferentially target NAc-projecting neurons. This is an interesting and important finding with significant implications for understanding projection-specific regulation of vHipp neural function in health and disease. There are a number of points to address to strengthen this manuscript.

The strategy for identifying mPFC- and NAc-projecting neurons in vivo is surprising and not entirely clear. The description in the methods along with the schematic in 4A suggests that fibre photometry was used to identify projection neurons based on retrograde expression of a myrGFP tag. Of note, the figure legend does not adequately describe this experiment or the associated traces in C and should be significantly expanded. However, piecing together the information throughout the methods and results suggests that an opto-electrophysiology probe was used, first using a blue light to excite the myrGFP in projection neurons, and then a yellow light to excite halorhodopsin in SST or PV cells, all the while recording cellular activity with an electrode. The problem with this approach is that fibre photometry is a population recording approach that does not afford single cell resolution and as such not an appropriate method for identifying individual projection cells. I have a number of additional questions about these experiments

- Were bursting neurons excluded?
- What was the rationale for excluding cells that had >20 % decrease in firing rate with laser excitation?
- In figure 4C, why is there a large jump in the fibre photometry signal?
- The experiment follows a repeated measures design, first recording baseline and then recording during halorhodopsin excitation. The data are reported as % change from baseline and appear to be normalized to the control group. It would be more transparent to simply show the firing rates for each group during baseline and stimulation periods and analyze following a repeated measures design.

It is not clear what the gene expression analyses of mPFC- and NAc-projection neurons add to this work. A similar analysis, cited in this manuscript, has already characterized differential expression in these and other vHipp projection neurons. For these results to make a significant contribution, more interpretation and integration is needed. Why is it interesting or important that these specific sets of genes show differential expression patterns between these projection pathways? Could this in some way inform our understanding of why PFC and SST interneurons show differential innervation of these populations? How much overlap is there between the present gene expression data and that of Gergues et al? Further, the cut-off for identifying differentially expressed genes is very liberal ($\log_2FC > 1$ with no correction for multiple testing). It also would seem surprising that, of the genes deemed to be differentially expressed, 92 of the 99 show upregulation in mPFC. Is this perhaps an

artefact? Were there any differences in RIN values or mapped read counts or other post-sequencing QC metrics?

REVIEWER COMMENTS

Reviewer #1 (Remarks to the Author):

The authors present some important work showing that projections from the ventral hippocampus (vHC) to the prefrontal cortex (mPFC) vs. to the nucleus accumbens (NAc) originate from distinct projection neurons spread across vCA1 and vSub; such distinct populations also show some differences in gene expression and - importantly - are differentially inhibited by SST-interneurons, while PV-interneurons innervate them both to roughly equal extent (although the variability within each population is extremely high, possibly explained by other factors not further analysed). The manuscript is very well written, the topic is very timely and the applied and combined methods are impressive and very state-of-the art. I have a few concerns, as follows:

Major Points

A. Fig. 1-2: major part of the discovered projection-specificity of certain vHC pyramidal neurons (PNs) seems to be mediated simply by a distinction between vCA1 and vSub according to Fig. 1C - and it was known before that the vHC targets the mPFC mainly from CA1 (Jay TM, Witter MP (1991), J Comp Neurol 313: 574–586) and the NAc mainly from the ventral subiculum (see e.g. studies from A. Grace and others), see also Bienkowski et al., Nat Neurosci, 2018 for a more recent and more detailed description; consequently, also the differential gene expression may be a majorly explained by differences between vCA1 and vSub PNs; is it possible to address this hypothesis with the data of the others and other existing gene expression datasets from vCA1 and vSub (e.g. from the Allen Institute, accessible through CytosploreViewer, or from the lab of Sascha Nelson)?

We agree that it is well known that anatomical subregions of the ventral hippocampus differentially target unique brain regions. We have now included the references listed above in the Discussion section. Importantly, it is also possible that the gene expression differences that we observed in these two pathways may be explained by differences across hippocampal subregions rather than differences between discrete projections. We have now performed an additional analysis of the differentially expressed genes using the Allen Brain Atlas Cytosplorer tool as suggested. As described in the Results and Discussion sections, only 12 of the 99 genes our RNASeq analysis identified were differentially expressed between the CA1 and subiculum of the hippocampus. This suggests that the differential gene expression is largely attributable to differences between hippocampal projections.

B. Fig. 4E / statistics: the analysis with a Kruskal-Wallis one-way ANOVA is unclear here; given that a parametric ANOVA was used with the rather non-parametric small-N distribution in 3C/F in order to be able to report interactions, it is unclear why with the more "parametric-looking"/high-N dataset in 4E no such ANOVA is used. Please calculate a univariate ANOVA (excluding controls) with the factors interneuron-type and projection-target to calculate the interaction between those two in order to be able to argue for a target-specificity of interneuronal connectivity; also, for statistical analysis of ratios, prior log-transformation should be considered; the one-way ANOVA is not wrong and can be done in addition, but given that the key claim of this dataset and the paper overall seems to be the target-specificity of interneuronal connectivity, one would really want to see a significant interaction to support this claim; if the claim is somewhat toned down to a differential connectivity of SST-interneurons as such (i.e. disregarding the connectivity-difference between PV and SST at this stage), then such a 2-way ANOVA and sign. interaction would not be necessary, but then one could even analyze PV and SST separately or only including controls in each analysis

We performed a non-parametric Kruskal-Wallis one-way ANOVA on the data presented in Figure 4E because it failed the Shapiro-Wilk normality test and therefore, the Two-ANOVA would be inappropriate. The data presented in Figure 3, however, passed the Shapiro-Wilk normality test, which is why we performed the parametric two-way ANOVAs. The results of the Shapiro-Wilk tests have now been added to the Results section to clarify why the specific statistical tests were used.

C. Fig. 4: a lot of the conclusion of this paper is grounded on the trust in the method of fluorescently guided in vivo recording of individual neurons used here, which in turn relies - somewhat blindly - on the suitability of the opto-electric probe to always record electrophysiological signals from the same neuron from which it also captures a fluorescence signal; is there any validation data (e.g. citable papers on this method) to support this experimental assumption (and possibly reports some error rates)? What about fluorescence light from nearby neurites of a recorded cell?

This technique was developed based on the original article published in Nature Methods (LeChasseur 2011). In this manuscript, the authors confirmed that the optical and electrical signals originated from the same cell by measuring the distance between the axial positions of the two signal peaks and only including cells in which the interpeak distance was less than the mean neuron diameter. Using this cut-off, 100% of the cells met the criteria. However, even when using more stringent cut-off (interpeak distance <90% of the average soma diameter), 75% of the recorded cells still met this criteria. This information has now been included in the methods section of the manuscript.

Minor Points

1. the authors write the Introduction as if a hyperactive vHC or vHCPFC connection would be clearly detrimental in the negative symptom domain, but neither integrate nor cite their own study demonstrating that the activation of this region/pathway exerts an anti-depressant response which even appears to be part of the long-term anti-depressant effect of ketamine (Carreno et al., Mol Psych, 2015); can the authors offer a more comprehensive view on the role of hyperactivity of this region/pathway for the negative symptom domain?

Thank you for bringing this point to our attention. We initially excluded our previous findings suggesting that the vHipp→mPFC, but not the vHipp→NAc, pathway regulates the antidepressant response to ketamine for the sake of simplicity as the primary message we were trying to convey is that the two pathways can have different effects on behavior. However, we understand that it seems counterintuitive that both activation and inhibition of the vHipp→mPFC can be beneficial, depending on the model we are examining. However, as with most systems in the brain, we expect the behavioral effects of vHipp →mPFC pathway activation to fall along an inverted U curve, where both too little and too much activation lead to abnormal behavior. We have now modified the Introduction section to cite our previous work.

2. p.5, please add details on the surgery, such as the analgesic regime used peri-operatively (aside from isoflurane anaesthesia), infusion speed and volumes, needle size, type and size of implanted cannula; also please detail further the opto-electrophysiological system used for data in Fig. 4 (e.g. individual components, order numbers, etc.)

We have now modified the methods section to include additional details related to the stereotaxic surgeries and the opto-electrophysiology set up.

3.p. 5 and later; please unify the writing of the greek "micro" in um

We apologize for this oversight. This has been corrected throughout the manuscript.

4. Fig. 2B, the gene names are not legible, please consider re-arranging Fig. 2 and enlarge the panel B; also the relevance of the identified genes presented here is not clear relative to the context of the physiological and behavioural differences of the two analysed projections - can anything more functionally relevant be drawn from this dataset, except for showing that the populations are different (e.g. two GPRs come up, but with uncertain neural relevance, another GPCR might be relevant, ...)?

We have rearranged Fig 2 in order to enlarge panel B. The primary finding presented in this figure is that multiple genes are differentially expressed in the two projection pathways. We believe that it is possible that the differential gene expression may explain the differences in synaptic connectivity, as multiple genes involved in synaptic pruning and plasticity were differentially expressed between the two pathways. Although testing this hypothesis is outside the scope of the current manuscript, we have added a sentence to the Discussion section to consider potential future directions.

5. Fig. 3 C, F, J ... the statistics displayed in the graph is confusing; in C and F one assumes that the asterisk indicates the post-hoc test within the SST-sub-groups, but in J the $p = 0.067$ seems to be what the authors cite as p-value of the interaction in the overall ANOVA; please indicate and/or state clearly the p-values of the subtype-target interaction, and of the pairwise post-hoc tests comparing target within SST and comparing subtype within mPFC-target (if relevant, seems to be different); also, given the small n-number (= mice, according to figure legend) and high variability in these essential panels, please state how many slices were analysed per mouse; apart from that, is there an explanation why the scale of the y-axes in C and F is so different, even though seemingly the same parameter is displayed? Final remark: the n-numbers are - understandably, given the used methods - rather small in these three panels which could be perceived as unconvincing, but the authors may want to strengthen the notion the three datasets clearly support each other, and are hence quite convincing as a set despite small n (just a suggestion)

We apologize for the confusion. We have now removed the $p = 0.067$ from the graph and more clearly labeled the post-hoc statistics to the Results section. In addition, we have now added more details regarding the n (number of cells/slices analyzed per mouse) to the Results and Figure legends. As far as the scale of y-axes, we believe that the discrepancy is a result of methodological differences. While eGRASP and confocal microscopy allows us to analyze a relatively large section of the apical dendrite in 3 dimensions, electron microscopy only provides a 2-dimensional image of a smaller portion of the dendrite. Therefore, we believe that the important finding is not the absolute number of synapses but rather the relative differences between specific circuits. Finally, we agree completely that although the total number of animals is not excessive, the conclusions drawn are supported by the fact that the same results were found across three unique techniques. We have amended the Discussion section to highlight this point.

6. Fig. 4C, it is unclear what is meant to be shown here and why a sudden (!) strong step in calcium influx occurs after ca. 2s of excitation - is the timeline the approach to the cell?; please describe more in detail, and possibly show more traces to illustrate the process

We apologize for the confusion and have now dropped "fiber photometry" from the manuscript and figure as it may be interpreted as the measurement of calcium. Instead, the opto-electric probe is used to identify fluorescently labeled cells. As we move through the brain, the opto-electric probe is used to shine blue light, which excites GFP fluorescence when we encounter a GFP+ cell. The emitted GFP fluorescence is captured by the same opto-electric probe and transmitted back through the optical fiber to a photomultiplier tube. The photomultiplier tube output is converted to a digital signal, which is shown in the bottom panel of Fig 4C. The goal of Fig 4C is to demonstrate that as we approach a cell, the firing rate increases (top panel) and there is an increase in emitted fluorescence (bottom panel). This has now been more clearly explained in the results and figure legend.

7. Fig. 4F, the text says traces "before and after optogenetic inhibition", but the figures seems to show traces before and during inhibition - which is correct? Which interneuron-type is inhibited here (and shown in 4G)?

We apologize for the lack of clarity. The text has been updated to say "before and during optogenetic inhibition." In addition, the results and figure legend now specify the exact treatment groups represented in the traces and images in Figs 4 F and G.

8. maybe the title is slightly misleading in that it sounds too general ("distinct microcircuits regulate") relative to the underlying data; the key point really is that NAc-projecting neurons are distinct from mPFC-projecting neurons (Fig. 1-2)

and that only the former get strongly inhibited by SST-interneurons (Fig. 3-4) - I am not requesting any change to the title, but just strongly suggest re-consider exchanging it for a more informative and accurate one; note that some of this over-generalization also appears in Abstract and Discussion, where it could be re-placed with more concrete statements

Thank you for this feedback. We have now modified the Title, Abstract, and Discussion sections to avoid overly generalized statements.

Reviewer #2 (Remarks to the Author):

The manuscript describes a study that seeks to establish the structural and functional relationship between PV and SST interneurons and distinct populations of ventral hippocampal projection neurons (NAc-projecting vs mPFC-projecting). The author's previous work has shown that transplantation of PV and SST into the ventral hippocampus may be a promising treatment for schizophrenia symptoms and the differential effects of PV vs. SST transplantation led to the main hypothesis that PV and SST interneurons are connected to distinct projection neuron subpopulations. The authors use a variety of viral tracing and physiological techniques to establish this differential connectivity. Although the manuscript is well-written, I have concerns about the quantification and analysis of the data and how that influences the interpretation of the results.

Major Concerns:

1) This study often discusses the ventral hippocampus (vHipp) as one solitary structure and the authors proceed to dissect the NAc-projecting vs. mPFC-projecting population. However, as the first dataset shows, the mPFC-projecting population is largely distributed within the CA1 whereas the NAc-projecting population is primarily in the ventral SUB. Because of this segregated distribution, it is unclear if the differences shown in the other experimental data are actually projection-specific (mPFC-projecting vs. NAc-projecting) or an effect of anatomical location (CA1 neurons vs SUB neurons). If the authors analyze CA1 and SUB groups within their data they could address this and strengthen their overall argument that these effects are projection-specific (ex. compare mPFC-projecting CA1 neurons vs mPFC-projecting SUB neurons). The CA1 and SUB are functionally distinct brain regions that should be treated independently.

We agree that the ventral hippocampus is not a solitary structure but can be divided into unique subregions. Further, it is possible that the gene expression differences that we observed between mPFC- and NAc-projecting neurons is a result of the anatomical location of the cells rather than the projection target. In an effort to address this comment, as well as the comment made by reviewer 1, we have now performed an additional analysis of the differentially expressed genes using the Allen Brain Atlas Cytosplorer tool. As described in the Results and Discussion sections, only 12 of the 99 genes our RNASeq analysis identified were differentially expressed between the CA1 and subiculum of the hippocampus. This suggests that the differential gene expression is largely attributable to differences between hippocampal projections rather than subregion differences.

2) The methodology described for the quantification of the experimental datasets is unclear. First, some characterization of the injection set size and location is critical for the interpretation of the results as this can affect the distribution of the retrogradely-labeled cells within the sampling area. As for the sampling area, it's unclear where the dorsoventral location of the horizontal section and if this is in fact sampling from the area with the most labeled cells (this should depend on the injection site locations). Overall, the sparse number of reportedly labeled cells (~5-6 cells per slice) suggests either the sampling area is insufficient or the AAV2retro viruses are not labeling the full population. Viral tropism with AAVretro viruses is an additional concern here as these viruses are known to weakly infect some cell types (ex. layer 6 cortical pyramidal cells). Non-viral tracers such as Fluorogold or Cholera toxin would be an alternative.

Figure 1 includes schematics to demonstrate the approximate injection sites and sampling area. Although we agree that AAV2retro may not label the full population of cells, we observed similar levels of labeling in the experiments that used AAV2retro and those that used Fluorogold (RNAScope), suggesting that the viruses are labeling a representative

population of cells. It is also important to note that in the eGRASP and electron microscopy experiments, we often observed labeling in many more than 5-6 cells per slice. However, we only analyzed a subset of the cells (3-4 per slice).

3) The RNAscope methodology states that the quantification was used to “determine the number of GFP-positive puncta per fluorogold-labeled cell” but the results are reported as “puncta per μm^2 ” which is not appropriate to truly determine if cells are differentially expressed. It is not clear how meaningful this effect is and how it compares to the differential expression demonstrated by the flow cytometry experiments. In summary, RNAscope data should be presented as number of puncta per cell and a comparison to flow cytometry would help to strengthen to two differential gene expression analyses.

We decided to analyze the RNAscope data as a function of the cell size, as the total soma area differed slightly from cell to cell. However, we have now also analyzed the data as total puncta per cell (regardless of size) and found the same result. We have now included this data in the Results section and Figure 2.

4) The eGRASP experiments also present the data as “synapses per μm^2 ” when this data should be presented as “synapses per cell” if the goal is to understand how SST and PV interneurons are differentially connected to different projection populations. Presenting the data like this would lead to overrepresentation of highly connected cells. Furthermore, the Two-way ANOVA does not say the effect is significant and the p value is not listed. P value and significance should be included and if the effect is not significant than it is not appropriate to perform post-hoc tests.

While the ability to analyze the total number of synapses per cell would be ideal, with the current methods, it is difficult to obtain an image of the full dendritic arbor and the analysis would be overly burdensome. Therefore, in the current experiment, we have opted to analyze discrete cellular compartments, including the proximal apical dendrite. Because there is some variation in the length of the dendrite analyzed, we believe it is more accurate to express the number of synapses as a function of dendrite length. Regarding the statistics, we apologize for the oversight. The Two-way ANOVA did indicate a significant interaction, and we have now included that information in the Results section.

5) The monosynaptic rabies tracing experiment should be analyzed by quantifying the number of SST and PV GFP positive cells as a ratio to the number of labeled “starter cells” within individual animals as the number of starter cells can vary between animals. The quantification as it currently is describes how PV and SST represent the percentage of total monosynaptic input but don’t help explain how many interneurons synapse on to an individual projection neuron cell type. Furthermore, judging by the graph in figure 3J, it appears that 5 of the 20 animals in these experiments didn’t have any monosynaptically connected interneurons labeled (0%).

The rabies experiments were performed in PV-Ai14 or SST-Ai14 mice so that the specific interneuron subtypes would be fluorescently labeled with RFP. Unfortunately, that meant that we had to use a herpes virus that did not include an mCherry tag. Therefore, we cannot differentiate the starter cells from monosynaptically connected cells because both cell types would express GFP. Therefore, we decided to express the data as the percentage of monosynaptically connected interneurons out of the total GFP positive cells, which we believe takes into account the variation in the number of starter cells. While we understand that this strategy does have some limitations, it is important to note that the findings corroborate those collected using different techniques (eGRASP and immunogold labeling/electron microscopy presented in Fig 3C and 3F, respectively). As suggested by reviewer 1, we have now added a sentence to the Discussion section to highlight this point.

Minor Concerns:

1) The methods sections states that quantification was performed on “2-4 sections per animal” but the number of animals is not listed in methods (only in figure caption). All of the sample size (n) should be reported in the methods section.

We have now updated the Methods section to include both the number of animals and the number of sections analyzed.

2) Figure 1, the fluorescent images are too small to discern the detailed labeling. I would suggest enlarging these images and minimizing the schematic.

We have now rearranged the picture so that the labeling can be discerned.

3) The 4% fluorogold concentration used in RNAscope experiments is likely too high. Greater than 2% is considered cytotoxic and likely caused necrosis at the injection site. This could effect the labeling of projection neuron populations in the vHipp.

We apologize for this mistake. We have updated the Methods section to indicated that 2% fluorogold was used.

4) Why are the number of synapses/ μm reported by the eGRASP experiments approximately 20-30 times higher than the synapses/ μm reported by the EM experiments?

We believe that the discrepancy is a result of methodological differences between the eGRASP and immunogold labeling techniques. While eGRASP and confocal microscopy allows us to analyze a relatively large section of the apical dendrite in 3 dimensions, electron microscopy only provides a 2-dimensional image of a smaller portion of the dendrite. It is likely that neither technique is labeling every single synapse; however, we believe that the important point is that the trends remain the same across the three techniques and we have now amended the Discussion section to point this out.

5) Figure 3D shows a representative eGRASP image but it is unclear from which experiment this image is from. Do PV and SST neurons synapse on to similar dendritic compartments for mPFC- and NAc-projecting populations? In addition, Imaris could be used to reconstruct the 3D morphology of the pyramidal cells and map the synaptic contacts to the dendrites.

We have now added information to the Results section and figure legend to indicate the experimental group where we observed this difference. In a subset of animals, we analyzed the number of synapses found in other cellular compartments (distal apical dendrite, cell body, basal dendrites). We did not observe obvious differences in the synaptic distribution and all cellular compartments examined seemed to show a similar trend (SST cells form fewer synapses on mPFC-projecting pyramidal cells).

6) Stereotactic coordinates are given for cannula placement but its unclear which stereotactic atlas was used for targeting. Are vHipp injections more in the CA1, SUB, or in between both?

We used the Paxinos and Franklin Brain Atlas, which we have now included in the References. The injection site was targeted at the intersection between the CA1 and SUB so that both subregions received the viruses.

Reviewer #3 (Remarks to the Author):

This manuscript explores an intriguing hypothesis that PV and SST interneurons in vHipp differentially innervate and regulate mPFC and NAc projecting neurons. This hypothesis is inspired by a finding that transplanting SST and PV neurons to vHipp differentially regulates behaviors associated with mPFC and NAc projections. Data from a variety of retrograde viral tracing strategies and in vivo electrophysiology with targeted optogenetic silencing of PV or SST interneurons demonstrate that, while PV interneurons innervate both mPFC and NAc projecting vHipp neurons, SST interneurons preferentially target NAc-projecting neurons. This is an interesting and important finding with significant

implications for understanding projection-specific regulation of vHipp neural function in health and disease. There are a number of points to address to strengthen this manuscript.

The strategy for identifying mPFC- and NAc-projecting neurons in vivo is surprising and not entirely clear. The description in the methods along with the schematic in 4A suggests that fibre photometry was used to identify projection neurons based on retrograde expression of a myrGFP tag. Of note, the figure legend does not adequately describe this experiment or the associated traces in C and should be significantly expanded. However, piecing together the information throughout the methods and results suggests that an opto-electrophysiology probe was used, first using a blue light to excite the myrGFP in projection neurons, and then a yellow light to excite halorhodopsin in SST or PV cells, all the while recording cellular activity with an electrode. The problem with this approach is that fibre photometry is a population recording approach that does not afford single cell resolution and as such not an appropriate method for identifying individual projection cells. I have a number of additional questions about these experiments

- Were bursting neurons excluded?

- What was the rationale for excluding cells that had >20 % decrease in firing rate with laser excitation?

- In figure 4C, why is there a large jump in the fibre photometry signal?

- The experiment follows a repeated measures design, first recording baseline and then recording during halorhodopsin excitation. The data are reported as % change from baseline and appear to be normalized to the control group. It would be more transparent to simply show the firing rates for each group during baseline and stimulation periods and analyze following a repeated measures design.

We apologize for the confusion surrounding the explanation of this technique, which was shared by multiple reviewers. We have now expanded the Methods section and figure legend to provide more clarity. In addition, we have now removed the term “fiber photometry” from the text and figure to avoid confusion. To answer the additional questions, no, we did not exclude bursting neurons. Our rationale for excluding cells that had >20% decrease in firing rate was that this effect was likely due to a polysynaptic response as optogenetically inhibiting interneurons should not decrease the activity of pyramidal cells if the interneurons are providing direct monosynaptic inhibition. In figure 4C, the electrophysiology and optical traces were taken as we approached a GFP-positive pyramidal cell. Therefore, the large jump in signal was observed because we moved next to a cell that was emitting fluorescence. This has now been clarified in the Results section and figure legend. Finally, we have now included a graph (Supp Fig 1) to show the firing rates during baseline and after stimulation.

It is not clear what the gene expression analyses of mPFC- and NAc-projection neurons add to this work. A similar analysis, cited in this manuscript, has already characterized differential expression in these and other vHipp projection neurons. For these results to make a significant contribution, more interpretation and integration is needed. Why is it interesting or important that these specific sets of genes show differential expression patterns between these projection pathways? Could this in some way inform our understanding of why PFC and SST interneurons show differential innervation of these populations? How much overlap is there between the present gene expression data and that of Gergues et al? Further, the cut-off for identifying differentially expressed genes is very liberal ($\log_2FC > 1$ with no correction for multiple testing). It also would seem surprising that, of the genes deemed to be differentially expressed, 92 of the 99 show upregulation in mPFC. Is this perhaps an artefact? Were there any differences in RIN values or mapped read counts or other post-sequencing QC metrics?

In response to this comment, and comments made by the first two reviewers, we have now performed an additional analysis. Specifically, we used a publicly available dataset (Allen Institute Cytosplorer Viewer) to determine whether the genes that we identified as differentially expressed between NAc- and mPFC-projecting cells were also differentially expressed across hippocampal subregions. This additional analysis suggests that the gene expression differences are a function of projection target rather than hippocampal subregion. Further, it is possible that the differentially expressed genes underlie the differences in connectivity. For example, mPFC-projecting pyramidal cells have higher expression of genes involved in synaptic pruning, which could offer a mechanism by which SST interneurons form fewer synaptic

connections on these cells. Although experiments to test this hypothesis are outside of the scope of the current manuscript, we have now added a sentence to the Discussion section to discuss this possibility and future directions. Finally, we did not observe differences in RIN values between samples and have now added this information, in addition to other quality control measures, to the Results section.

REVIEWER COMMENTS

Reviewer #1 (Remarks to the Author):

The authors have addressed all my concerns adequately and significantly improved their manuscript. In my view, the manuscript is ready for publication in Nature Communications. D. Kätzel

Reviewer #2 (Remarks to the Author):

The revised manuscript has addressed some of the previous issues, but several major concerns about the data analysis remain that inhibit the interpretation of this dataset.

1. A new gene expression analysis was reported using the "Allen Brain Atlas Cytosplorer" tool. Appropriate reference to this tool's website needs to be made in the manuscript as I could not find this through a Google search. I may be wrong, but I think the tool is called "Cytosplore Viewer" and is developed by a team at TU Delft and LUMC using data from the Allen Cell Types Database generated by Allen Institute and other labs that contributed to the Brain Initiative Cell Census Network (BICCN) found here <https://viewer.cytosplore.org>.

2. The Cytosplore data shows nonsignificant difference between CA1 and SUB for the Grn and Nedd9 genes, but also that Grn expression is 4 times higher than Nedd9 in both CA1 and SUB. Regardless of the anatomical location, this would suggest Grn expression would be higher than Nedd9 expression. However, the representative images shown in Fig 2F don't seem to reflect this difference. Do the authors find support for this in their quantification datasets?

3. The cartoon schematic in Figure 1 is insufficient to characterize the injection site size and specificity which is critical to interpreting the cell-type specific analysis. Particularly, your tissue is sectioned horizontally and your schematic is showing a coronal section, so I don't know how you can correctly estimate the size and placement. Please include representative microscopy images of the AAVretro and Fluorogold injection sites as part of a supplementary figure. Injection site size and placement is a fundamental first-step toward any anatomical tracing analysis.

4. For the RNAscope analysis, it is said that the total number of puncta per cell was analyzed, but no count is reported other than a Two-way ANOVA (but also no post-hoc Holm Sidak is listed). Please report the mean puncta per cell and include it in Figure 2. Also, a reference to representative images in Figure 2G is made, but there is no panel G in Figure 2.

5. The results of the rabies experiment were not significantly different based on the two-way ANOVA ($p=0.067$). The way this section is written is misleading and the statement in the discussion that "this observation was made in 3 separate experiments using 3 distinct anatomical techniques (i.e. eGRASP, immunogold labeling and electron microscopy, and monosynaptic rabies tracing)." Is not supported by the results.

6. The inability to characterize starter cells in monosynaptic rabies tracing experiments inhibits the interpretation of these results. Furthermore, there is no way to characterize the accuracy of the Herpes virus injections and with a system that relies on two injection sites to be accurately placed, there can be a lot of variability in the amount of starter cell populations between cases.

Typo error in Figure 2F caption

As mentioned before Fig. 2G is missing. Maybe this was a mistake in uploading the revised figure.

Reviewer #3 (Remarks to the Author):

The revised manuscript addresses many of the concerns raised in the first round of reviews. The addition of further methodological details regarding the opto-electrophysiological recordings is helpful and greatly clarifies the experiment addressing the majority of critiques related to this experiment. However, I still have concerns about the robustness of the conclusions drawn from this technique. This is not yet a widely used technique. Including some kind of validation that the technique is working as published within the present experiment is essential to support interpretation of the data presented. The representative traces in figure 4C look vastly different from both the description (see below), graphics (Fig 1D) and representative traces (Fig 3D) provided in the original paper (LeChasseur et al, 2011) that described this method.

'With this design, fluorescence collection is asymmetrical, with only light in front of the tip being collected. Therefore, as the probe moves past a labeled neuron, the detected fluorescence signal should follow a characteristic intensity profile: growing gradually when approaching a labeled cell and dropping abruptly when the probe moves beyond the cell (Fig. 1d).'

Figure 1D in the manuscript shows an abrupt step in the fluorescence signal. The original work suggests a gradual increase as the probe moves towards a labeled neuron. Can the authors explain why in their preparation they observe a sharp, discrete increase?

Reviewer #1 (Remarks to the Author):

The authors have addressed all my concerns adequately and significantly improved their manuscript. In my view, the manuscript is ready for publication in Nature Communications.

Reviewer #2 (Remarks to the Author):

The revised manuscript has addressed some of the previous issues, but several major concerns about the data analysis remain that inhibit the interpretation of this dataset.

1. A new gene expression analysis was reported using the “Allen Brain Atlas Cytosplorer” tool. Appropriate reference to this tool’s website needs to be made in the manuscript as I could not find this through a Google search. I may be wrong, but I think the tool is called “Cytosplore Viewer” and is developed by a team at TU Delft and LUMC using data from the Allen Cell Types Database generated by Allen Institute and other labs that contributed to the Brain Initiative Cell Census Network (BICCN) found here <https://viewer.cytosplore.org>.

We apologize for this oversight. The references to this tool have been corrected throughout the manuscript.

2. The Cytosplore data shows nonsignificant difference between CA1 and SUB for the Grn and Nedd9 genes, but also that Grn expression is 4 times higher than Nedd9 in both CA1 and SUB. Regardless of the anatomical location, this would suggest Grn expression would be higher than Nedd9 expression. However, the representative images shown in Fig 2F don’t seem to reflect this difference. Do the authors find support for this in their quantification datasets?

Our RNA Sequencing data also indicate much higher Grn expression when compared to Nedd9. The RNAScope experiments, however, do not show obvious differences between overall Grn and Nedd9 expression. We believe that this discrepancy could be related to differences between the two techniques. For example, RNA Scope is dependent on the hybridization efficiency of each probe. In addition, in the RNAScope experiments, we only measured and analyzed mRNA expression in the cell body (as fluorogold predominately labels perikarya). In our own RNA Sequencing data and in the Cytosplore Viewer dataset, it is possible that dendritic expression of these two genes (which are known to affect synaptic function) contributed to overall expression level.

3. The cartoon schematic in Figure 1 is insufficient to characterize the injection site size and specificity which is critical to interpreting the cell-type specific analysis. Particularly, your tissue is sectioned horizontally and your schematic is showing a coronal section, so I don’t know how you can correctly estimate the size and placement. Please include representative microscopy images of the AAVretro and Fluorogold injection sites as part of a supplementary figure. Injection site size and placement is a fundamental first-step toward any anatomical tracing analysis.

We apologize for the confusion. In experiment depicted in Figure 1, a coronal slice was made at the level of the optic chiasm. The anterior portion of the brain was sliced coronally to determine injection sites which the posterior portion was sliced horizontally to allow analysis of hippocampal subregions. This has been clarified in the text. In addition, a representative image showing the size and placement of mPFC and NAc injections have been added to Figure 1.

4. For the RNAScope analysis, it is said that the total number of puncta per cell was analyzed, but no count is reported other than a Two-way ANOVA (but also no post-hoc Holm Sidak is listed). Please report the mean puncta per cell and include it in Figure 2. Also, a reference to representative images in Figure 2G is made, but there is no panel G in Figure 2.

We apologize for the oversight. The values for puncta per cell have been added to the results. In addition, the graph showing puncta per cell is now included in Fig 2F, and representative images are now shown in Fig 2G. Post-hoc analysis was not performed as only a main effect of projection target was observed.

5. The results of the rabies experiment were not significantly different based on the two-way ANOVA ($p=0.067$). The way this section is written is misleading and the statement in the discussion that “this observation was made in 3 separate experiments using 3 distinct anatomical techniques (i.e. eGRASP, immunogold labeling and electron microscopy, and monosynaptic rabies tracing).” Is not supported by the results.

We have edited the statement to indicate that the observation was made using eGRASP and electron microscopy to avoid misleading the reader.

6. The inability to characterize starter cells in monosynaptic rabies tracing experiments inhibits the interpretation of these results. Furthermore, there is no way to characterize the accuracy of the Herpes virus injections and with a system that relies on two injection sites to be accurately places, there can be a lot of variability in the amount of starter cell populations between cases.

We agree that this is a limitation of this experiment and have added this information in the manuscript. However, it is important to note that the conclusions drawn in the paper do not rely on this technique alone. Importantly, both the eGRASP and electron microscopy experiments found similar results.

Typo error in Figure 2F caption

This typo has been corrected.

As mentioned before Fig. 2G is missing. Maybe this was a mistake in uploading the revised figure.

We apologize for this mistake. The uploaded Figure 2 now includes panel G.

Reviewer #3 (Remarks to the Author):

The revised manuscript addresses many of the concerns raised in the first round of reviews. The addition of further methodological details regarding the opto-electrophysiological recordings is helpful and greatly clarifies the experiment addressing the majority of critiques related to this experiment. However, I still have concerns about the robustness of the conclusions drawn from this technique. This is not yet a widely used technique. Including some kind of validation that the technique is working as published within the present experiment is essential to support interpretation of the data presented. The representative traces in figure 4C look vastly different from both the description (see below), graphics (Fig 1D) and representative traces (Fig 3D) provided in the original paper (LeChasseur et al, 2011) that described this method. ‘With this design, fluorescence collection is asymmetrical, with only light in front of the tip being collected. Therefore, as the probe moves past a labeled neuron, the detected fluorescence signal should follow a characteristic intensity profile: growing gradually when approaching a labeled cell and dropping abruptly when the probe moves beyond the cell (Fig. 1d).’ Figure 1D in the manuscript shows an abrupt step in the fluorescence signal. The original work suggests a gradual increase as the probe moves towards a labeled neuron. Can the authors explain why in their preparation they observe a sharp, discrete increase?

The fluorescence signal is affected by multiple factors that could explain why we observed a sharp increase in fluorescence rather than the more gradual increase observed by LeChasseur. Specifically, the speed at which the electrode is moving throughout the brain likely affects the rate of change of the signal. Indeed, LeChasseur reports (Fig 4e) that signal intensity in brain is observed over a distance of $\sim 20\mu\text{m}$ and, in our hands, extracellular recordings electrodes are typically advanced in $\sim 10\mu\text{m}$ increments resulting in sharp increases in intensity. In addition, differences in methodological parameters such as optoelectrode composition, photomultiplier tube sensitivity, GFP expression, or laser intensity may also contribute to differences in the dynamics of the fluorescent signal.

REVIEWER COMMENTS

Reviewer #2 (Remarks to the Author):

The authors have updated the revised manuscript and satisfied all of my concerns.

Reviewer #3 (Remarks to the Author):

As requested in the previous round of reviews, some kind of technical validation for the in vivo imaging technique is essential to support this data. This has not been provided.

As the authors note, their results look different to the original publication. This is not a standard or widely used technique. Validate that it works.

Reviewer #2 (Remarks to the Author):

The authors have updated the revised manuscript and satisfied all of my concerns.

Reviewer #3 (Remarks to the Author):

As requested in the previous round of reviews, some kind of technical validation for the in vivo imaging technique is essential to support this data. This has not been provided.

As the authors note, their results look different to the original publication. This is not a standard or widely used technique. Validate that it works.

We have now performed an additional experiment to validate the opto-electrophysiology technique. Specifically, the red shifted Channel Rhodopsin, C1V1, and YFP were expressed in pyramidal cells of the hippocampus. We have now included electrophysiology traces to show an increase in action potential magnitude and fluorescent signal as we move toward a fluorescently-labeled cell along the dorsal-ventral axis (Fig 4B). Importantly, only fluorescently labeled cells, which express C1V1, increase their firing rate in response to yellow laser stimulation, whereas the firing rate of spontaneously active neurons that are not fluorescence are unaltered (Fig 4D). It should be noted that this experiment was performed in a different laboratory than the original experiments, which explains why the opto-electrophysiology set up was modified slightly (including different lasers and a new micro-positioner). In a previous review, one concern related to the sharp increase in fluorescence that was observed as we approached a cell. We have now included a trace that shows a more gradual increase in fluorescence, which resulted from a slower movement along the dorsal-ventral axis, made possible with the new micro-positioner.

REVIEWERS' COMMENTS

Reviewer #3 (Remarks to the Author):

The authors have addressed my comments and the manuscript is now suitable for publication.

REVIEWERS' COMMENTS

Reviewer #3 (Remarks to the Author):

The authors have addressed my comments and the manuscript is now suitable for publication.

Thank you for the useful feedback. We are glad that the additional experiment addressed your previous concerns.